# Bioactive Glasses and Glass-Ceramics for Healthcare Applications in Bone Regeneration and Tissue Engineering

**DOI:** 10.3390/ma11122530

**Published:** 2018-12-12

**Authors:** Hugo R. Fernandes, Anuraag Gaddam, Avito Rebelo, Daniela Brazete, George E. Stan, José M. F. Ferreira

**Affiliations:** 1Department of Materials and Ceramic Engineering, CICECO, University of Aveiro, 3810-193 Aveiro, Portugal; h.r.fernandes@ua.pt (H.R.F.); anuraagg@ua.pt (A.G.); avitorebelo@ua.pt (A.R.); d.s.b@ua.pt (D.B.); 2National Institute of Materials Physics, RO-077125 Magurele, Romania; george_stan@infim.ro

**Keywords:** bioactive glasses, alkali-free, scaffolds fabrication, additive manufacturing techniques, bone regeneration, tissue engineering

## Abstract

The discovery of bioactive glasses (BGs) in the late 1960s by Larry Hench et al. was driven by the need for implant materials with an ability to bond to living tissues, which were intended to replace inert metal and plastic implants that were not well tolerated by the body. Among a number of tested compositions, the one that later became designated by the well-known trademark of 45S5 Bioglass^®^ excelled in its ability to bond to bone and soft tissues. Bonding to living tissues was mediated through the formation of an interfacial bone-like hydroxyapatite layer when the bioglass was put in contact with biological fluids in vivo. This feature represented a remarkable milestone, and has inspired many other investigations aiming at further exploring the in vitro and in vivo performances of this and other related BG compositions. This paradigmatic example of a target-oriented research is certainly one of the most valuable contributions that one can learn from Larry Hench. Such a goal-oriented approach needs to be continuously stimulated, aiming at finding out better performing materials to overcome the limitations of the existing ones, including the 45S5 Bioglass^®^. Its well-known that its main limitations include: (i) the high pH environment that is created by its high sodium content could turn it cytotoxic; (ii) and the poor sintering ability makes the fabrication of porous three-dimensional (3D) scaffolds difficult. All of these relevant features strongly depend on a number of interrelated factors that need to be well compromised. The selected chemical composition strongly determines the glass structure, the biocompatibility, the degradation rate, and the ease of processing (scaffolds fabrication and sintering). This manuscript presents a first general appraisal of the scientific output in the interrelated areas of bioactive glasses and glass-ceramics, scaffolds, implant coatings, and tissue engineering. Then, it gives an overview of the critical issues that need to be considered when developing bioactive glasses for healthcare applications. The aim is to provide knowledge-based tools towards guiding young researchers in the design of new bioactive glass compositions, taking into account the desired functional properties.

## 1. Background: Nature, Structure, and Chemistry of Glasses

### 1.1. The Nature of Glasses

Glasses have been used by mankind for thousands of years in multiple forms and applications. Their overall importance in the everyday life might not be fully realised at a first glance due to the vast variety of appliances in fields ranging from simple and common materials (e.g., bottles, windows, containers, light bulbs, etc.) to technical applications (e.g., television tubes, computer screens, spectacles and telescope lenses, spectrometer prisms, laboratory ware, optical fibres, etc.) or artistic purposes [1,2,3].

Unlike crystalline solids, glasses do not show any long-range order or any significant symmetry in their atomic arrangement [4]. Instead, the atoms constituting the glass are organised in a short-range order that depends on the glass composition, i.e., glass seems to be a liquid, but behaves similar to a solid in human time scales [4,5]. While glass appears as a rigid solid material, at geological time scales, it is actually relaxing towards the supercooled liquid state. Therefore, a glass structure is unstable with respect to the supercooled liquid state, and the supercooled liquid is metastable with respect to the equilibrium crystal [6]. The structure of glasses directly affects many glass properties, such as glass stability, density, coefficient of thermal expansion (CTE), solubility, and ion release in aqueous environments. The eventual occurrence of surface modifications upon reacting with natural or synthetic physiological fluids is regarded as a signal of bioactivity.

Glasses are usually processed through two principal methods: melt-quenching and sol-gel processes. In the melt-quenching process, the batch is heated at high temperatures (usually >1300 °C) and quenched to freeze the disordered atomic structure of the melt to get an amorphous glass. In most cases, the glass is heat-treated at lower temperatures in order to relax the thermomechanical stresses in the structure induced by rapid cooling. The sol-gel process uses inorganic or organic precursors, which are subjected to different processes of hydrolysis and condensation, followed by drying and thermal stabilisation heat treatments; through this process, glass can be shaped into powder, nanoparticles, fibres, coatings, etc.

In terms of compositional flexibility, glasses have great advantages in comparison to crystalline substances. While crystalline phases possess well-defined and constant stoichiometry, glasses can be synthesised into a virtually unlimited range of compositions, which can be designed to adjust to a specific necessity [5]. Moreover, the properties of a glass can be adjusted by doping, i.e., adding small amounts of other oxides to the glass composition, allowing the controlled release of ionic species. This feature is of great importance in bioactive glasses, conferring them with potential therapeutic actions upon releasing suitable ions that might stimulate cells’ differentiation (osteoinduction), or act as antimicrobial or neuroprotective agents, etc. [7]. 

Other important difference between glasses and crystalline materials is the existence of the glass transition temperature (*T_g_*) in the former, at which the solid glass becomes a viscous liquid glass. More correctly, *T_g_* is an interval of temperatures that depends significantly on the composition and thermal history of the material (e.g., melting temperature, cooling rate, and the subsequent heat-treatment schedules).

### 1.2. Glass Structure

One of the earliest consideration of the glass structure was proposed by Zachariasen in his classic paper [8]. Today, his ideas remain central to the field of glass research; they are called *random network theory*. According to Zachariasen, the atoms in a crystal and a glass are linked together by identical interactions and vibrate about their equilibrium positions. However, the main structural distinction between both is that a glass lacks periodicity and symmetry in the structure, in contrast to a crystal. Due to lack of symmetry, the properties of glasses are isotropic (unless prepared in an external field). Another essential consequence of the lack of symmetry is that the unit cell of the glass is of infinite size. Next in his paper, Zachariasen went into considerable detail on the glass structure by taking examples of oxide glasses. In analysing the structure of vitreous silica, Zachariasen noticed that the glass network is built up of oxygen tetrahedra surrounding silicon atoms. The tetrahedra are connected to each other by corner sharing, such that each oxygen atom is linked to two silicon atoms. A two-dimensional illustration of this structure is presented in Figure 1a,b, with tetrahedra represented as triangles. Zachariasen concluded that a vitreous network can only be built by oxygen tetrahedra or oxygen triangles, because oxygen octahedra or oxygen cubes would lead to periodic structures.

Zachariasen noticed that the following general rules hold for vitreous oxides.
(1)Each oxygen atom is linked to no more than two cations.(2)The oxygen coordination number of the network cation is small.(3)Oxygen tetrahedra or triangles share only corners, and not edges or faces.(4)At least three corners of each oxygen polyhedron must be shared in order to form a three-dimensional (3D) network.(5)Sample contains a high percentage of cations, which are surrounded by oxygen tetrahedra or by oxygen triangles.

These general rules have become rules for glass formation. However, they do not explain the formation of glasses in non-oxide systems, and some of the rules are not valid, even for the oxide systems: for example the existence of oxygen triclusters [9]. Therefore, these rules should not be used dogmatically; however, they still provide a starting point to the structural analysis of glasses.

By introducing some components such as alkali and alkaline earth oxides into the vitreous oxide networks, the extra oxygens do not form bridges, but rather form free ends, as shown in Figure 1c; these have a different structural functionality. Therefore, depending upon the type of structural function, the components in oxide glasses are divided into three groups:(1)*Network formers* are the components that build the glass network by forming oxygen tetrahedra and oxygen triangles, which are also called *network units* or *structural units*. The network formers are the essential components of the glass, being able to form 3D structures. These units are connected to each other by corner sharing creating oxygen bridges, which are called *bridging oxygens* (*BO*), as described by random network theory (shown in Figure 1a). The common examples are SiO_2_, B_2_O_3_, and P_2_O_5_. Their schematic representations are presented in Figure 2.(2)*Network modifiers* are the components that break down the glass network by creating terminal oxygens, which are also called *non-bridging oxygens* (*NBO*), as shown in Figure 1c. The common examples are alkali and alkaline earth oxides.(3)*Intermediate oxides* are the components that assume either the role of network formers or network modifiers, depending on glass composition. Some of the most common examples are MgO and Al_2_O_3_ (which is often not welcome in bioactive glasses, because contents greater than 1.5 wt.% Al_2_O_3_ in the glass tend to turn it bioinert, inhibiting bone bonding [10]).

In common glass production, monovalent oxides such as Na_2_O are added to act as fluxing agents, decreasing considerably the melting temperatures of the glasses, and therefore reducing the production costs. However, binary compositions of alkali silicate glasses have low chemical durability. For this reason, the oxides of divalent cations, such as CaO, are added to stabilize the glass [4]. Common window glass is usually based on the soda–lime–silica (Na_2_O–CaO–SiO_2_) system [11]. In the case of most bioactive glasses, the base components are usually SiO_2_, Na_2_O, CaO, and P_2_O_5_ [12].

Among the numerous possibilities of cataloguing glasses, they can be classified according to their compositional system. Usually, this takes into consideration the network former oxide(s) that are present in the composition. Therefore, glasses can contain just one network former oxide, such as silicate, phosphate, and borate glasses, or more complex compositions with mixed glass network former oxides such as borosilicate, phosphosilicate, or borophosphate glasses [13].

#### 1.2.1. Silicate Glasses

Silicon dioxide (SiO_2_), a common component of sand, is the most common network former due to the high valence of silicon (Si^4+^). The basic building unit of silicate glasses is the SiO_4_ tetrahedron (Figure 2), which can be connected to up to a maximum of four other tetrahedra through covalent bonds (≡Si–O–Si≡) via its corners [14]. Adding network modifiers to glass composition results in the disruption of the continuity of the glassy network due to the cleavage of some of the ≡Si–O–Si≡ bonds, leading to the formation of non-bridging oxygen (*NBO*) groups and resulting in a decrease in the number of *BO* and network connectivity (*NC*) [15]. The different kinds of silicate structures that are formed are commonly described using the *Q^n^* notation, where *n* represents the number of bridging oxygens per tetrahedron, and varies between zero and four [4,16]. This notation will be used and detailed in the next section, which is dedicated to glass structure.

#### 1.2.2. Phosphate Glasses

Phosphate glasses consist of an inorganic phosphate network of PO_4_^3−^ tetrahedral units (Figure 2), with each one connected to a maximum of three other phosphate tetrahedral units through covalent ≡P–O–P≡ bonds [17]. The structure of P retains its fourfold coordination throughout the full composition range from pure P_2_O_5_ to orthophosphates fully saturated with alkali oxides [18]. Structurally, the effect of the incorporation of network modifier oxides into phosphate glasses is similar to the effect in silicate glasses, i.e., the P–O–P bonds are broken, and *NBO* atoms are formed. When these glasses contain more modifiers than phosphate, they are often called invert glasses, and the glass properties are dominated by ionic bonds between *NBO* and modifier cations, rather than the covalent P–O–P bonds [14,19]. 

#### 1.2.3. Borate Glasses

B_2_O_3_ is one of the most important glass-forming oxides due to its higher field strength, lower cation size, small heat of fusion, and trivalent nature of B. It has been used in numerous glass systems in order to get specific properties [20]. High amounts of network-modifying oxides can be incorporated into borate glasses, making them interesting matrix materials for various applications [21,22]. Vitreous B_2_O_3_ consists of planar BØ_3_ trigonal groups (with Ø corresponding to *BO* atoms) predominantly grouped in six-membered boroxol rings (Figure 2). Adding M_2_O modifier oxides to B_2_O_3_ results in the transformation of neutral BØ_3_ trigonals into negatively charged [BØ_4_]^−^ tetrahedral (Figure 2), which are charge balanced by the modifier cation *R* [22,23]. Therefore, in contrast to phosphate or silicate glass, the initial addition of modifier cations to B_2_O_3_ increases the *NC*. For higher modifier oxide additions, the tetrahedral borate units convert back to trigonal borate groups, increasing the number of *NBO* and decreasing the *NC*. The non-linear change in *NC* with modifier content in borate glasses explains the compositional dependence of glass properties. This phenomenon is commonly named the ‘boron anomaly’ [22,24,25].

#### 1.2.4. Mixed Glass Former Systems

Many glass compositions are based on more than one former oxide, such as borosilicates, borophosphates, or phosphosilicates, for instance. The combination of several network formers permits obtaining glasses with specific medleys of physical properties [26]. Some silicate glasses contain B_2_O_3_ or P_2_O_5_ in their compositions, but they are not usually considered to be mixed-glass former systems, because the amounts of these oxides are small, or they do not participate as oxide formers in the glass network. Borosilicate glasses are among the most important technical glass systems due to their interesting properties, being used in applications ranging from laboratory glassware to optical glasses. In some bioactive glasses, SiO_2_ was partially to fully replaced by B_2_O_3_ [27]. Ternary alkali borophosphate glasses are known for their remarkable physical and chemical stabilities, and enhanced ionic conductivities, which make them suitable for applications as electrolytes for solid-state batteries [28]. Borophosphate glasses have also been suggested for specific applications such as biomedical uses (e.g., degradable temporary implants) [29,30]. Boroaluminosilicate glasses can find applications in diverse fields such as bioactive materials for the regeneration of bone and tissue [31], radioactive waste containment [32], or fiberglass for composite applications [33].

### 1.3. Probing the Structure

The structure of a glass could be quantified by the pair distribution function (PDF) or radial distribution function (RDF), given by *g*(*r*), which is related to the probability of finding another atom at a distance *r* from a central atom. The PDF is generated by Fourier transformation of the structure factor *S*(*Q*) that is acquired from X-ray, neutron, and electron diffraction techniques. The PDF describes the short-range structure, and it could be universally applied to all material systems, namely solids, liquids, and gasses. However, in the case of oxide glasses, a better way of quantifying the short-range structure would be by *Q^n^* distribution, which is also called *Q^n^* or network speciation. The *Q* represents a network-forming polyhedron, while *n* is the number of bridging oxygens (*BO*) that are associated with it. A glass composition under given equilibrium conditions has its network polyhedra distributed among several *Q^n^* units, hence the name ‘*Q^n^* distribution’. The *Q^n^* notation has become an extremely useful tool for the structural analysis of glasses. As crystallography is for the research of crystalline materials, so is *Q^n^* distribution for oxide glass research. This *Q^n^* distribution in a glass composition can be directly accessed by experimental techniques such as nuclear magnetic resonance (NMR) and Raman spectroscopy [34,35,36]. NMR spectroscopy is especially apt for determining the *Q^n^* distribution, because it probes the short-range structure around a nucleus under examination.

In silicate glasses, depending upon the number of *BO*s and *NBO*s present on a particular silicate tetrahedron, the unit is called a *Q^n^* unit; where *n* is the number of *BO*s, and *n* ∈ {0, 1, 2, 3 and 4}. Other network formers such as B_2_O_3_ and P_2_O_5_ also undergo network speciation when added into silicate glasses. Similar to SiO_2_, P_2_O_5_ also exists as a tetrahedral unit, having one doubly-bonded oxygen on one corner, which acts as terminal oxygen, similar to *NBO*. On the other three corners, the oxygens can be either *BO* or *NBO*. Therefore, depending upon the number of *BO*s and *NBO*s present on a particular phosphate tetrahedron, the unit is called a *Q^n^*^(*P*)^ unit; where *n* is the number of *BO*s, and *n* ∈ {0, 1, 2 and 3}. Furthermore, B_2_O_3_ in borate and borosilicate glasses undergoes a different kind of network speciation, where it speciates into three coordinated trigonal units (*B^III^*) and four coordinated tetrahedral units (*B^IV^*). The *B^IV^* has a net one unit of negative charge on it; therefore, in order to balance the charge, it requires a cation called a *charge compensator*. This is fulfilled by network modifiers that also act as charge compensators. The individual boron units, just as silicate units, could further speciate in terms of the number of *BO*s and *NBO*s on each unit forming different extended structures [37]. The boron speciation in borosilicate glasses is described by empirical models proposed based on the experimental data [38,39,40,41]. 

Modelling the *Q^n^* distribution is extremely important in studying oxide glasses. Therefore, over the last few decades, several theoretical models have been proposed to describe the *Q^n^* distribution in binary silicate glass systems [42,43,44]. However, these models have several limitations, and therefore, a statistical mechanical model has been proposed to describe *Q^n^* distribution in multicomponent oxide glasses with respect to temperature and composition [45,46].

## 2. Glass-Ceramics: Between Glasses and Crystalline Materials

Glass-ceramics (GCs) were discovered in the mid-20th century, and can be considered as a combination of a glass with a ceramic [47]. GCs derived from the controlled crystallisation of parent glasses, resulting in material containing one or more crystal phases embedded in a residual glass, depending on the starting glass composition and the given heat treatment [48]. The molecular rearrangements that occur during the crystallisation process allow producing the crystalline structures, which are often metastable polymorphs that can be transformed to the thermodynamically stable crystal phases under additional and appropriate heat treatment [48]. The choice of parent glass composition and the controlled nucleation and crystallisation allow producing GCs with a wide variety of special microstructures; most of them would be impossible to produce in any other kind of material. Further, the crystal phases developed during the crystallisation possess an even wider variety of characteristics [47,48].

Glass-ceramic materials possess two remarkable advantages over other types of materials. The first is that GCs can be obtained in highly complex shapes, because they are derived from a parent glass that can be moulded by using a variety of well-known inexpensive techniques (e.g., casting, pressing, rolling, etc.) [49,50]. The second great advantage is that GCs usually consist of a very fine microstructure containing few or no residual pores, resulting in improved mechanical properties. GCs feature interesting combinations of properties, including biological, electrical, thermal, and mechanical [51,52,53].

The selection of glass composition and the heat treatment schedule are key factors for GC processing. The components of the glass and their proportions should promote the precipitation of specific crystalline phase (or phases), avoiding long heat treatments in order to get the desired final properties. Some glass compositions feature very high thermal stability, and are very difficult to crystallise, e.g., 13–93 [54], while others feature very narrow sintering windows and readily crystallise above *T_g_*, such as 45S5, resulting in undesirable microstructures [24]. The success of controlled crystallisation also depends on the heat treatment, including the heating rate, nucleation temperature, crystallisation temperature, and the heating time at each specific temperature. These factors are fundamental for converting the glass into homogeneous microcrystalline GCs featuring desirable finely uniform or interlocking microstructures in order to achieve good mechanical properties.

An effective crystallisation should initiate with the production of a large number of small crystals, rather than a small numbers of large crystals [55,56]. Therefore, the nucleation stage is an important step in the heat treatment process. Nucleation is followed by the crystal growth upon the developed nuclei at higher temperature. The heating rate must be carefully controlled in order to avoid the deformation of the GC during the process. Good selection of the heating rate, crystallisation temperature, and time result in effective crystallisation and decreasing of the remaining glassy phase.

Most of the times, the heat treatment of the parent glass converts the transparent material into a translucent or opaque polycrystalline GC [57]. This effect is caused by the scattering of the light at the interfaces between adjacent crystals, and between the crystals and the residual glass phase due to differences in the refractive index [57]. When the developed crystals are very small (near or smaller than the wavelength of light) and the crystalline phase and glass have close refractive index values, the resultant GC can appear to be transparent or slightly translucent. This is an important feature for materials that require specific aesthetic properties such as GCs that are used as dental materials [47,57].

## 3. Bioactive Glasses and Glass-Ceramics

The progressive aging of world population, coupled with an increasing incidence of skeletal diseases, is a main driving force stimulating the increasing research efforts put forward developing new implantable materials. The same factors are behind parallel determinations for improving the manufacturing techniques to fabricate implant devices aiming at regenerating and repairing living tissues damaged by disease or trauma [58]. The discovery of bioactive glasses in the late 1960s was triggered by a challenging discussion in 1967 between Larry L. Hench and Colonel Klinker on a bus ride towards the United States (U.S.) Army Materials Research Conference held in Sagamore, New York [31]. Larry L. Hench was an assistant professor at the University of Florida, and Colonel Klinker had had recently returned to the United States from Vietnam, where had served as an Army medical supply officer. There were big concerns with the number of amputations derived from the body’s rejection of inert metal and plastic implants, and an urgent need for a novel material that could form a strong bond with living tissues. Soon after (1968), a project proposal with this focused aim was funded by the U.S. Army Medical Research and Design Command in 1968 [31]. In the following year (1969), the bonding ability of a glass composition (45SiO_2_–24.5Na_2_O–24.5CaO–6P_2_O_5_ wt.%) to bone and muscle after six weeks post-implantation in rats was firstly observed [31,59,60]. This glass composition was then trademarked by the University of Florida as 45S5 Bioglass^®^. Its discovery changed the paradigm in the biomaterials field, shifting the interest from bioinert to bioactive materials. Such an important finding stimulated interest in pushing this bioactive glass towards the market. Such endeavours culminated with the approval by Food and Drug Administration (FDA) of the first 45S5 Bioglass^®^ implant (MEP^®^ implant for middle ear ossicular repair) in 1985 [61,62]. 

The initial idea of Larry Hench was to combine elements that are abundant in the human body in proportions that favour the rapid initial dissolution of alkalis from the glass surface in aqueous solutions, followed by the precipitation of a Ca-rich and P-rich layer at the inner alkali-depleted silica layer [63]. The Na_2_O–CaO–SiO_2_ system was selected, to which P_2_O_5_ was also added in small amounts. In fact, when such glasses are immersed in biological fluids, a layer of hydroxyapatite [HA, Ca_10_(PO_4_)(OH)_2_] or hydroxyl carbonated apatite (HCA) similar to the mineral phase of bone is formed on the glass surface. The bone-bonding ability is conferred by this HCA layer, which is chemically and structurally similar to the mineral apatite phase found in bone tissue. The formation of an HCA layer occurs according to the followed sequence of reactions, as proposed by Hench [64]:(1)The alkali and alkali earth ions are firstly released into the fluid and are replaced in the glass structure by H^+^ or H_3_O^+^ ions present in the fluid, which leads to an increase in the local pH, causing the rupture of the Si–O–Si bonds.(2)The disruption of the glass network permits the releasing of silicon into the fluid in the form of silanol Si(OH)_4_ groups.(3)For local pH values lower than 9.5, Si(OH)_4_ groups condensate, and form a polymerized silica gel layer on the surface of the glass.(4)The open structure of silica gel allows the continuity of ionic exchange between the glass and the fluid: the Ca^2+^ and P^5+^ ions diffused from the glass, together with the Ca^2+^ and P^5+^ ions present in the fluid, form an amorphous calcium phosphate layer over the silica gel.(5)The amorphous calcium phosphate layer incorporates carbonate species and begins to crystallise into HCA.

Bioactive glasses commonly exhibit faster rates of HCA formation and bone-bonding formation [10]. For instance, bioactive glasses show a higher osteoconductivity than bioactive ceramics, because the osteoconductivity is associated with the formation of HCA, and the rate of the superficial HCA formation of bioactive glasses is higher than that for ceramics. These materials undergo degradation over time when in contact with body fluids, being gradually replaced by new bone formation and tissue regeneration. This discovery of bioactive glasses as the first artificial materials with a demonstrated ability to form an integrated bond with bone and soft tissues stimulated much interest from scientists and clinicians [60]. New applications beyond middle ear ossicular repair [62,65] were envisaged, with 45S5 Bioglass^®^ also being marketed in particulate form under the trade name of Perioglas^®^, which was used to fill periodontal bone defects, and more recently as injectable pastes and putties under the trade name of NovaBone^®^. The chronology of the key applications in biomedicine of these and other bioactive glasses is well summarised in a recent publication [66]. According to another recent report, 45S5 Bioglass^®^ has been clinically applied in more than 1.5 million patients [61]. 

Unfortunately, this bioactive glass composition presents several shortcomings related to its high alkali content. These include: (1)relatively fast dissolution and resorption rates [67] that negatively affect the balance of natural bone remodelling, and lead to gap formation between the tissue and the implant material [68];(2)poor thermal properties with a close proximity between glass transition temperature (*T_g_* ~ 550 °C) and the onset of crystallisation (*T_c_* ~ 610 °C), hindering densification and resulting in weak mechanical strength and early crystallisation [69,70,71,72,73,74]. This represents a serious limitation when envisaging the manufacture of highly porous scaffolds.(3)its high coefficient of thermal expansion precludes using it as adherent coating material for metallic, polymeric, or ceramic implants [75];(4)the high pH value created by the high doses of sodium leached to the culture medium cause serious cytotoxic effects [76,77] reported on a series of glasses compositions expressed as (16.20 − *x*)MgO − *x*Na_2_O–37.14CaO–3.62P_2_O_5_–42.46SiO_2_–0.58CaF_2_ (in mol %, with *x* varying between zero and 10) prepared by the melt quenching technique. They found that increasing sodium contents at the expenses of MgO induced: (i) an increase in cytotoxicity towards the mouse-derived pre-osteoblastic MC3T3-E1 cell line; (ii) a delayed formation of HCA surface layer upon immersing in simulated body fluid (SBF) solution; (iii) a slight depolymerization trend in the silicate glass network, accompanied by an enhanced affinity of alkali cations towards phosphate.

Cannillo et al. [78] found that a prolonged contact of polycaprolactone (PCL)-45S5 Bioglass^®^ composite scaffolds with water, used to leach the pore former salt, suppressed the development of hydroxyapatite in vitro, with calcite being preferentially formed. The concomitant leaching of sodium mitigated the cytotoxicity response. Fabbri et al. [79] also prepared polycaprolactone (PCL)-45S5 Bioglass^®^ (BG) composites with BG weight contents varying in the range 0–50% using a solid–liquid phase separation method (SLPS). Either dimethyl carbonate (DMC) or dioxane (DIOX) was used as solvent, and ethanol was used as the extracting medium. The ability of the composites to induce the precipitation of hydroxyapatite increased with increasing BG contents. The poor in vitro performance that was observed for cytotoxicity and osteoblast proliferation tests were attributed to the poor wettability of the composites, but the effects of sodium leaching cannot be discarded. Alginate dialdehyde (ADA), gelatin (GEL), and nanoscaled 45S5 Bioglass^®^ (nBG) hydrogel films were investigated, aiming at compromising biocompatibility, cellular adhesion, proliferation and differentiation properties, and obtaining predictable degradation rates. The in vitro evaluation allegedly showed good cell adherence and proliferation of bone marrow-derived mesenchymal stem cells seeded on covalently cross-linked alginate dialdehyde-gelatin (ADA-GEL) hydrogel films with nBG. However, significant increases in lactate dehydrogenase (LDH) and mitochondrial activity were observed for both ADA-GEL and ADA-GEL-nBG groups compared to alginate. Such increases are signs of cytotoxicity, as this enzyme is released by cells upon their death. Meanwhile, an augmented mitochondrial activity would advocate for a good cell proliferation, and thus these coupled observations seem to be contradictory. However, the addition of even a very small content of nBG (0.1%) caused a slight cytotoxic effect compared to ADA-GEL [80]. Meng et al. developed Bioglass^®^-based scaffolds loaded with tetracycline microspheres for drug delivery in bone tissue engineering. They concluded that the constructs exerted limited cytotoxicity in mouse fibroblast cells [81]. Ball et al. [82] performed toxicity assessments using mouse osteoblasts cultured directly on porous scaffolds made of ceria and of 45S5 Bioglass^®^ for 72 h, and concluded that the toxicity of ceria was rather lower in comparison to that exerted by 45S5 Bioglass^®^. Composite films of gelatin–starch (GS)—bioactive glass 45S5 microparticles (m-BG) loaded with vancomycin hydrochloride (VC) were investigated to evaluate the degradation in vitro, the release profile of VC and its antistaphylococcal effect, and cytotoxicity in MG-63 osteoblast-like cell cultures [83]. The liquid extracts from the composites resulted in cytotoxic effects. These last effects were attributed to the presence of glutaraldehyde (GA), which was used as the cross-linker (0.25 wt.%), but the main cause of the observed cytotoxic effects was probably the sodium leaching, as the addition of 45S5 to GS films led to increasing weight losses over time due to the dissolution of m-BG.

Alno et al. [84] used a modified hanging drops method to generate spheroids with a well-established human fetal osteoblasts line (hFOB 1.19) to study the effect of 45S5 Bioglass^®^ ionic dissolution products in comparison with two-dimensional (2D) cultures. They observed cell enhanced proliferation in 2D conditioned cultures in comparison to 3D cultures in conditioned spheroids under the same testing conditions, without a change of gene expression patterns. Based on an apparent agreement with clinical observations showing the insufficiency of commercially available bioglasses for bone repairing within nonbearing sites, the authors suggest that this model could be adapted for the screening of innovative bioactive materials by laboratory techniques that are already available and the extended monitoring of their bioactivity. However, no load bearing was used in both 2D and 3D cultures in the present study. 

Zhang et al. [85] compared the cytocompatibility of two kinds of porous bioactive glass-ceramics: an apatite–wollastonite (A-W) bioactive glass-ceramic prepared by the sol-gel method, and 45S5 Bioglass^®^ prepared by the melt quenching method, using bone marrow stromal cells (BMSCs). The viability of BMSCs cultivated with the extraction of these two kinds of biomaterials was investigated. The extract of 45S5 had significantly higher cytotoxicity. Accordingly, the amount of cells that adhered to A-W and exhibited good bioactivity and the cytocompatibility was superior in comparison to 45S5.

Aiming at enhancing the understanding the role of piezoelectric barium titanate (BT), which is often explored in orthopaedic research to electrically stimulate bone-forming cells, Ball et al. [86] investigated the in vitro biocompatibility of porous scaffolds made of BT or of 45S5 Bioglass^®^ using a mouse osteoblast (7F2) cell line. The cytotoxicity exerted by 45S5 Bioglass^®^ after 72 h was higher (8.4 ± 1.5%) in comparison to that of BT (6.4 ± 0.8%). This lower in vitro performance of 45S5 Bioglass^®^ is highly concerning, considering the barium muscle poison effect that is expected from the non-stoichiometric dissolution of BT [87]. Therefore, besides the pH increase due to the ionic exchange involving the dissolution of BT, the leached Ba^2+^ will cause added toxic effects. Barium directly stimulates all types of muscles, including cardiac muscle, and causes a profound reduction in serum potassium together with an increase in intracellular potassium [88]. As a matter of fact, it is this highly toxic effect of barium that justifies the use of BaCO_3_ as an effective rat poison, causing weakness and hypokalaemia [89]. 

Adopting a preconditioning (sodium leaching) step is defended as a smart strategy to reduce the posterior ion exchange reaction and the associated pH rise in the culture medium that causes cytotoxicity in the surrounding environment [90]. The glass is immersed in either simulated body fluid (SBF) or culture medium for a pre-determined time period. Using this approach, Pryce and Hench [90] preconditioned two types of bioactive glass particulates (melt quenched derived 45S5 Bioglass^®^ and sol-gel synthesized 58S) in SBF for periods ranging from 30 minutes to 48 h, and evaluated the effects on glass dissolution and subsequent HCA formation. The subsequent calcium dissolution in culture medium was reduced by 20 ppm for 45S5 Bioglass^®^ and by 75 ppm for 58S, while sodium release from 45S5 Bioglass^®^ was also diminished by 100 ppm. Based on Fourier transform infrared (FTIR) spectroscopy data, the authors did not find significant changes in the HCA layer formation rate when using either of the glass powders (non-leached and sodium leached), concluding that the preconditioning step does not adversely influence bioactivity. However, this is just re-centring the focus on a secondary and misleading aspect, as the concluding remarks would be centred on the cytotoxicity effects. However, this sodium-leaching approach just gives illusory results, since the same preconditioning step is not practicable in the operation room.

Based on the plethora of existing evidence about the cytotoxicity of 45S5 Bioglass^®^ and the other bioactive glass compositions inspired from it, the declared absence of cytotoxic effects exerted by 45S5 Bioglass^®^-derived glass-ceramic scaffolds [91], or particles in amounts within the range of 0–200 μg mL^−1^ [92], or sintered GC pellets prepared from sol-gel derived powders [93], is rather questionable unless a preconditioning step has been used. However, its description is constantly omitted in these literature reports.

The interest in bioactive glasses has been continuously increasing, as will be reviewed below in Section 3.1. Several research works were aimed at further exploring the properties of 45S5 Bioglass^®^ in an attempt to expand the potential applications and overcome some of its main drawbacks. The cytotoxicity effects are also common to other glass compositions inspired by the 45S5 Bioglass^®^ [94]. The effects of replacing CaO by SrO in the 45S5 Bioglass^®^ composition in 50% (Sr50) or 100% (Sr100) on a molar basis on density, solubility, and in vitro cytotoxicity were studied. Strontium substitution increased the solubility and caused statistically significant reductions in cell viability. 

Attempting to escape the in vitro cytotoxicity and the proneness of 45S5 Bioglass^®^ for crystallisation, the processing ability and properties of 45S5 Bioglass^®^ were compared with those of a modified glass (S520) with the nominal composition: 52.0SiO_2_–18.0CaO–20.9Na_2_O–2.0P_2_O_5_–7.1K_2_O (mol %) [95]. 3D scaffolds were produced by rapid prototyping using CO_2_ laser cladding without using moulds. Extensive crystallisation occurred when 45S5 Bioglass^®^ was used, while limited surface crystallisation was reported for S520 due to an expanded sintering window. Contrarily to 45S5 Bioglass^®^, the S520 scaffolds were not cytotoxic in vitro when osteoblast-like MC3T3-E1 cells were cultured with the dissolution products of the glasses.

Other 45S5 Bioglass^®^-derived formulations (mol %) with high Ca contents (BG_Ca/Mix: 2.3Na_2_O–2.3K_2_O–45.6CaO–2.6P_2_O_5_–47.2SiO_2_ and BG_Ca/K: 4.6K_2_O–45.6CaO–2.6P_2_O_5_–47.2SiO_2_) and a low tendency to devitrify were synthesised by sol-gel (SG) [96,97]. The cytotoxicity tests performed with murine fibroblasts (BALB/3T3) showed that the obtained SG glasses and their extracts did not allegedly induce negative effects on cell viability [98].

Investigating the relation between the in vitro cytocompatibility of SiO_2_–ZnO–CaO–SrO–Na_2_O glass compositions and their ion release profiles, Towler et al. [99] concluded that these compositions show equivalent or enhanced in vitro compatibility in comparison to the commercially available bioactive glass Novabone^®^ [99].

Another important concern with respect to bioactive glass porous scaffolds (vastly 45S5-based) [73,100,101,102,103,104,105,106,107] is their precarious overall mechanical performance and brittleness, which would make them unsuitable for real clinical applications. During the last decade, a facile, yet efficacious approach has been advanced to mitigate this technological limitation, namely the infiltration/filing or coating of the scaffolds porous structure with resorbable biocompatible polymeric materials. This solution is nature-inspired, as the cancellous bone is a complex organic–inorganic composite material having as its main components collagen (ossein) and multi-substituted HCA [100,101]. Moreover, cancellous bone serves as a model for the architecture of scaffolds. However, since Philippart et al. [101] have recently thoroughly reviewed this specific topic, we shall restrict ourselves to only mention here the most important polymeric materials used for this purpose. The polymeric materials employed in conjunction with bioactive glass-based bone scaffolds can be either *synthetical* (e.g., polyvinyl alcohol (PVA) [102], poly-DL-lactic acid [108], poly(ε-caprolactone) [103,109], poly(lactic-coglycolic acid) [110], poly(d,l-lactide)/poly(ethylene glycol)-(polypropylene glycol)-poly(ethyleneglycol) [104]), *natural* (e.g., collagen [105], bacteria-derived poly-(3-hydroxybutyrate) [73] or poly(3-hydroxybutyrate-co-3-hydroxyvalerate) [106], chitosan [111], cellulose [107], alginate [112], silk [113], zein [103], or gelatin [100]) or *hybrid* (poly(ε-caprolactone)/zein [103]; PVA/microfibrillated cellulose [102]) in origin. The usage of gelatin, as obtained by the temperature or chemical processing of collagen, can alleviate the risk of antigenicity, which is typically associated with animal-origin compounds [100]. Overall, the polymer–bioglass research studies can still be considered as being in their incipient development stage [101]. Definite designs that are able to improve the mechanical properties of the scaffolds while enabling drug-release capabilities, without radically altering the bioactivity, surface energy/wettability, or degradability of the products, are much desired and expected in the near future.

### 3.1. Bioactive Glasses in Numbers

The number of papers published per year in the field has noticeably increased especially since the beginning of the 1990s. The main goal of this section is to present a brief survey of the global bibliography output in the field of biomaterials collected from the available sources of information. A particular emphasis was given to bioactive glasses as scaffolds materials for tissue engineering and health applications. Besides the sources of information, the conditions under which data were gathered are also presented. In a first approach, a relatively broad (B) SCOPUS^®^ search was performed based on the following keywords: scaffold AND biomaterial appearing in the title, abstract, and keywords (TITLE-ABS-KEY) within the timeline window from 1970–2020. This search (B search) resulted in 10,515 documents. All of the information was downloaded in comma-separated values (CSV) file format for further processing ease. The first task was to discriminate the total number of publications by access type (open access, other). The resulting outputs are summarized in Table 1. It can be seen that the open access publications account for less than <5% of the total [114]. The same timeline window (1970–2020) was considered in both searches.

Considering the total huge number of publications (NP) gathered, another SCOPUS^®^ search (A search) better addressing the intended research activities under the focus of the present review was carried out with the intention of reducing the scope of information. The set of keywords that was selected for this search A was: scaffolds AND materials AND tissue AND engineering AND bioglass, also appearing in the title, abstract, and keywords (TITLE-ABS-KEY). This new search resulted in a total of 264 documents. Both searches were carried on 17 July 2018 in the SCOPUS^®^ website [114]. Figure 3 plots the NP against time for both queries A and B.

The lower time line limit is justified by the discovery of bioactive glasses by Larry Hench in late 1969 [59,60]. Table 2 and Table 3 indicate the respective most relevant publication areas for both queries A and B. It can be seen that the NP was not more than one per year in the years 1972, 1983, 1984, 1986, 1987, and 1991, with the exception of 1989 scoring four publications. The entrance of 45S5 into the market in the mid-1980s [61,62,65,66] stimulated the interest of researchers in bioactive glasses and their biomedical applications, with an upsurge of the related publications per year reaching numbers between five and seven in mid-1990s. After that, there was a fast and continuous increase in the NP per year, reaching more than 40 in 2000, and about 700 in 2010. 

A relative stagnation in the NP per year apparently occurred up to 2013, followed by a jump to more than 1000 within 2014–2017. However, in comparison with the first decade, the overall increasing rate along the second decade of this 21st century was smaller. The initial stagnation in the beginning of the second decade of the 21st century, and the overall deceleration in the NP per year, are certainly reflexes of the resilient negative effects brought by the worldwide economic crisis that erupted in 2008. The overall funding allocated to research activities drastically decreased in extent and at rates that certainly varied from one country or region to another, although a more specific analysis and detailed discriminations by country or region are beyond the scope of this review. 

The main subject publication areas corresponding to query A are reported in Table 2. It can be seen that the subject areas with the largest number of hits are ‘Materials Science’, ‘Engineering’, and ‘Biochemistry, Genetics, and Molecular Biology’. It is worth mentioning that a single paper in this interdisciplinary field of research usually includes more than one subject area, explaining why the sum of the items in all of the areas significantly exceeds the total NP.

The information gathered according to query B, and summarised in Table 3, reveals a similar scenario concerning the most relevant subject areas of publication. The four subject areas with largest number of hits are ‘Engineering’, ‘Materials Science’, ‘Biochemistry, Genetics, and Molecular Biology’, and ‘Medicine’. The subset of information gathered according to the more restrictive query A shows that the first reports are dated from 2000, with two items published (Figure 2). In the subsequent years, there was an increasing trend that is almost parallel to that observed for query B. 

Considering that all the data refer to 17 July 2018, the current year is expected to end up with a higher NP in both series. A paper of the B series with the publication date of 2019 is already available. 

Table 4 sorts out the data gathered in both research queries from the highest to the lowest values by the type of document (article, conference paper, review, and other type of documents—such as book chapter, editorial, notes, articles in press, short surveys, letters, amongst others). In both cases, the overall percentage of articles, conference papers, and reviews was always above 95% (96.59% for A query and 95.36% for B query). 

From now onwards, only the A query will be considered, since it is the one that better fits the topic of this review paper. From this perspective, Table 5 provides information about the main publication ‘Source Titles’ (ST) and the respective NP. 

Since the total NP per ST (NPST) was too high (110), the information was organised in a quantisation bucketing form (or data binning) namely, in subsets of ST having the same NP. The obtained sets after the bucketing were {1, 2, 3, 4, 5, 13, 17, 21, 24, 25}. From Table 5, one can infer that subsets with bucketing greater than or equal to three correspond to 59.09% of the total publications (TNP = NPST × ST), and this group corresponds to 20 ST for over a total of 110 ST for the A query. 

The ‘Journal of Biomedical Materials Research Part A’ appears in the first position, with 9.47% of the TNP; and the ‘Journal of Materials Science: Materials in Medicine’ appears in the second place, with 9.09% over the TNP. In the third, fourth, and fifth positions appear ‘Acta Biomaterialia’ (7.95%), ‘Biomaterials’ (6.44%), and ‘Materials Science and Engineering C’ (4.92%).

The results of the A query are sorted out in Table 6, according to the ‘Affiliation’ of the authors and the NP from the highest to the lowest values. After the bucketing of information, the following subsets were obtained {1, 2, 3, 4, 5, 6, 7, 8, 10, 11, 12, 13, 15, 16, 29, 67}. The total number of affiliations at the moment of the A query search was 160, combined in a total of 506 manners (meaning that many of these publications are resulting from collaborative works involving researchers from different institutions; remember that the TNP is 264 for the A query—please, see Table 1).

It can be seen that European institutions are clearly leading the research in this multidisciplinary field (positions 1–3 and 6–7, among others), followed by Chinese institutions (positions 4–5 and 8, among others). The leading positions of the Imperial College of London and of the Friedrich-Alexander-Universität Erlangen-Nürnberg are not much surprising, considering that their most active researchers within their PhD and post-doctoral work programmes have been conducted in the sleigh of the Christmas father Larry Hench [67,115,116,117,118], and could slide faster and smoother. However, the NP by one institution in a given country is also strongly dependent on many other factors, including the size of the population, the percentage of the population dedicated to research activities, the overall funds and facilities made available for carrying out the projects, the research policies, among several others. 

Gathering detailed data about the role of each factor is cumbersome. Therefore, our analysis will be restricted to the total number of population by country and the respective gross domestic product (GDP). The data were obtained from the United Nations (UN), Department of Economics and Social Affairs [119] (under the section downloads, which presents freely data uploaded in December of 2017, the search was carried during July of 2018). Finally, but not the least important, care was taken to organise and present the information by each country, and the respective number of publications. Table 7 presents the NP by each country along the period under analysis (2000 up to July of 2018, A query), alongside with the 2017 population and respective 2016 GDP (breakdown at current prices in US dollars for all of the countries and regions. The reason why the 2016 GDP values are used is because they are the most recent ones available online at the moment of the search).

According to the ‘NP/Million Habitants’ ratio reported in Table 8, the first, second, third and fourth positions belong to Belgium, the United Kingdom, Portugal, and Finland, respectively. Regarding the NP/GDP ratio, the first, second, third, and fourth positions are occupied by Hong Kong, Serbia, Taiwan, and Belgium, respectively.

Thus, it is important to look at the data gathered from different perspectives in order to devise some tentative conclusions. It seems that GDP is the factor that most affects the positioning in the sorted ranks, followed by the number of people in each country. Unfortunately, it was not possible to get more detailed information about how many people in each country are engaged in this very specific field of research, as well as about the budget allocated by each country to research projects related to the A query (scaffolds, materials, tissue, engineering and bioglass), or even going deeper at the level of budgets per institution, which should shed further light on the affiliation relevance. It would be very interesting gathering and treating this type of detailed information in the future. However, such analysis will represent a heavy task, considering that databases for data mining often require their own analytic engine, design, and structure, and are not always publically available/accessible. 

However, the global bibliography output perspective provided in this section is just a way of looking into the research activities related to the specific areas under focus in this work, being far from conveying an overall and complete picture of the state of the art in the field of bioactive glasses and glass ceramics intended for healthcare applications. As a matter of fact, presenting the global bibliography output in numbers does not provide any critical and insightful appraisal in terms of relevance and the specific contribution of each published work to the state of the art.

Most of the reported bioactive glass compositions investigated so far [65,66,118,120,121,122,123,124,125,126,127,128,129] were inspired by the 45S5 Bioglass^®^, and contain significant amounts of alkali oxides (Na_2_O, K_2_O) that decrease the melting temperature of the glass. However, as previously stated, they can significantly reduce their in vitro and in vivo performances due to the sudden release of alkali ions, as reported above. On the other hand, the confirmed mismatch between the high rates of dissolution and degradation of 45S5 Bioglass^®^ and the growth of new bone in rabbits [130,131] compromise bone regeneration, especially in defects with critical size. This justifies an insightful revision on the relationships between glass structure, dissolution, and bioactivity.

In terms of NP per million habitants, the United Kingdom appears in the first position, followed by China in the second position, Germany in the third position, and the United States of America in the fourth position. 

Sorting out the data based on the ratios of ‘Number of publications/Million Habitants’ and ‘Number of publications/GDP in 2012 USD’, gives the lists presented in Table 8. The results of this exercise are interesting and confirm the repositioning of each country.

### 3.2. Glass Structure, Dissolution Behaviour, and Bioactivity

The chemical durability of glass is a crucial property for bioactivity, because the dissolution rate must be compatible with the cellular processes and with the rate of new bone formation. The dissolution of the glass in contact with the body fluids and the release of calcium and phosphate ions are crucial to the biomineralisation of the glass. Therefore, the dissolution behaviour plays a key role in the bioactivity of a glass. Bone BG grafts are expected to provide temporary support and enhanced tissue regeneration and growth [12,64]. If the dissolution rate is too slow, the ionic concentrations are insufficient to stimulate cellular proliferation and differentiation; on the contrary, if the dissolution rate is fast, the ionic concentrations might be beyond the effective level. The bonding rate of bioactivity glasses to bone tissues strongly depends on the glass composition as firstly demonstrated by the pioneering work of Larry Hench et al. [12], allowing the classification the materials in two categories: *Class A*—undergo rapid surface reactions, resulting in osteoconduction and osteostimulation; and *Class B*—undergo slower surface reactions and insignificant ionic release, conferring only osteoconduction properties, i.e., the ability of new bone migration along the implant interface. According to this criterion, the *Class A* compositions featured faster bone growth and more new bone formed in a graft site than the *Class B* ones. 

The dissolution and the ion release kinetics depend on the glass network structure and the type of ions present in the glass. Highly polymerised glass networks (larger *n* values of *Q^n^*) are slow to dissolve, and vice versa. In the field of bioactive glass research, the 10993-14 standard of the International Organization for Standardization (ISO) is usually employed to study the dissolution behaviour and its dependence on network structure as given by a parameter called *network connectivity* (*NC_old_*). This parameter describes the degree of the network polymerisation [132], and can be used to predict the dissolution extent and bioactivity. *NC_old_* is the average number of *BO* per network-forming polyhedron in a glass. In other words, it gives the average value of *n* in a *Q^n^* unit. However, such predictions become more difficult in the presence of mixed network former units, since the network modifiers are unequally distributed among the different types of network formers. Therefore, a *modified network connectivity* (*NC*) parameter was introduced, assuming that all of the phosphate species exist as orthophosphate (PO_4_^3−^) units, and *NC* only represents silicate connectivity [133]. This was considered to be a reasonable assumption in glasses with only P and Si network formers and relatively low P contents. For the optimal balance between the dissolution and degradation of bioactive glasses, *NC* = 2 is preferred. Edén [134] developed a generalised mathematical framework for the distribution of modifiers among multiple network formers that was called *Split Network Theory* (SNT). However, in either of the cases, the distribution of the modifiers among different network formers must be determined *a priori*, and cannot provide its dependence on temperature and composition. The influence of the glass network structure in defining the dissolution and bioactivity was studied for phosphosilicate alkali-free bioactive glass compositions co-doped with different molar levels of Zn^2+^ and Sr^2+^ [135]. The results revealed that there was no straightforward correlation between the *NC* and the dissolution behaviour. The dissolution was especially affected by the specific chemistry of ionic species in the glass, including valence and ionic radii, which determine their corresponding leaching behaviours [135]. Although the *NC* values of these co-doped glasses are slightly lower in comparison to that of Bioglass^®^ 45S5, their dissolution rates were slower. Similar results were obtained in yttrium containing soda lime phosphate glasses. Investigations from molecular dynamic simulations of bioactive glasses also showed that *NC* does not capture other important aspects of structural features such as the network-modifying cation clustering and modifier–chain bonding [136].

Research studies from other fields within glass science also face similar challenges when dealing with multiple network formers. Modelling the modifiers distribution as a function of temperature and composition is essential for the design and understanding of new glasses and GCs. Therefore, several research studies relied on the statistical mechanics for the modelling of this distribution [46,137,138]. The bioactive glass research could greatly benefit from these statistical mechanical models. In some bioactive glasses with only P and Si network formers [135], ^31^P NMR investigation reveals some substantial amount of pyrophosphate (P_2_O_7_)^4−^ evidencing the network speciation. The statistical mechanical model developed in our group [46] also accounts for the network speciation of each split network along with modifier distribution.

Due to these interesting features, bioactive glasses are considered third generation biomaterials, as they have the ability to induce specific intrinsic cell responses while bonding to hard tissue as well as to soft tissue [139]. Silicate bioactive glasses feature remarkable osteoconductive and osteoinductive properties, including the ability to promote angiogenesis and bond to both hard and soft tissues [140,141,142]. Such attractive properties make them appropriate for a large variety of clinical applications, from orthopaedic and dental fields to soft-tissue restoration and wound healing [31,65,143,144]. The dissolution products can stimulate cell behaviour due to their capability of releasing biologically active ions when in contact with body fluids and tissues, and degrade over time permitting bone regeneration. Moreover, bioactive glasses possess an amorphous structure, allowing the incorporation/releasing of specific ions with therapeutic effects [65]. 

### 3.3. Thermodynamics and Kinetics of Dissolution

In contrast to glasses for most other applications, which are expected to have low dissolutions rates to minimise the corrosion, bioactive glasses require specific dissolution rates to tune the in vitro and in vivo performances. The glass dissolution involves the breakdown of the glass network structure in the leaching medium. The dissolution behaviour of glasses depends on the thermodynamic activity of its components in the dissolution medium [145]. Moreover, factors such as (i) surface area of the glass/volume of the leaching solution and (ii) the type of dissolution medium (open/closed system) also affect the dissolution rates.

The glass dissolution behaviour is described based on the thermodynamics of hydration of the different components in the glass. It consists of three processes: (1) ion exchange, (2) matrix dissolution, and (3) back precipitation [146]. The ion exchange process involves the exchange of alkali/alkaline earth ions from the sample with H^+^ ions from the solution. While during the matrix dissolution, the siloxane-like bonds are broken due to the reaction with hydroxyl groups, resulting in the hydration of silica. The reactions occurring at the glass surface–solution interface can be summarised by the following equations:(R1)$$\equiv \mathrm{Si}-\mathrm{OM}+H_{2}O\;\Leftrightarrow \;\equiv \mathrm{Si}-\mathrm{OH}+M^{+}+{\mathrm{OH}}^{-}$$
(R2)$$\equiv \mathrm{Si}-\mathrm{Si}\equiv \left(\mathrm{glass}\right)+{\;\mathrm{OH}}^{-}\Leftrightarrow \;\equiv \mathrm{Si}-\mathrm{OH}\;+\equiv \mathrm{Si}-O^{-}$$
(R3)$$\equiv \mathrm{Si}-O^{-}\;+{\;H}_{2}O\;\Leftrightarrow \;\equiv \mathrm{Si}-\mathrm{OH}\;+{\;\mathrm{OH}}^{-}$$

The reaction (R1) creates a depleted alkali/alkaline ion zone on the glass surface. Further, the hydrated silica (≡Si–OH) results in the formation of a gel zone. The formation of a gel zone sometimes acts as a barrier leading to the retardation of the dissolution of the glass. Figure 4 shows a schematic of different zones. 

Apart from the ion exchange and matrix dissolution, there will be the back precipitation of the compounds (Figure 4), resulting in the formation of an amorphous layer or crystalline products. In the case of Wollastonite [145], the above set of reactions could be written as follows:(R4)$${\mathrm{CaSiO}}_{3}\left(\mathrm{glass}\right)+2H^{+}\left(\mathrm{aq}\right)\overset{K_{1}}{\Leftrightarrow }H_{2}{\mathrm{SiO}}_{3}\left(\mathrm{glass}\right)+{\mathrm{Ca}}^{2+}\left(\mathrm{aq}\right)$$
(R5)$$H_{2}{\mathrm{SiO}}_{3}\left(\mathrm{glass}\right)\overset{K_{2}}{\Leftrightarrow }{\;H}^{+}\left(\mathrm{aq}\right)+{\mathrm{HSiO}}_{3}{}^{-}\left(\mathrm{aq}\right)$$

These hydration reactions are reversible first-order equilibrium reactions with an associated equilibrium thermodynamic Gibbs free energy (ΔG_i_). The equilibrium constants for reactions (R4) and (R5) can be written as:(1)$$K_{1}=\frac{a_{H_{2}{\mathrm{SiO}}_{3}}.a_{{\mathrm{Ca}}^{2+}}}{a_{{\mathrm{CaSiO}}_{3}}.a_{H^{+}}{}^{2}}$$
(2)$$K_{2}=\frac{a_{\mathrm{HSi}O_{3}^{-}}\cdot {\;a}_{H^{+}}}{a_{H_{2}{\mathrm{SiO}}_{3}}}$$

Therefore:(3)$$\log K_{1}=-\frac{\Delta G_{1}}{\mathrm{RT}}=\log\left(a_{H_{2}{\mathrm{SiO}}_{3}}\right)+\log\left(a_{{\mathrm{Ca}}^{2+}}\right)-\log\left(a_{{\mathrm{CaSiO}}_{3}}\right)+2\mathrm{pH}$$
(4)$$\log K_{2}=-\frac{\Delta G_{2}}{\mathrm{RT}}=\log\left(a_{{\mathrm{HSiO}}_{3}{}^{-}}\right)-\mathrm{pH}-\log\left(a_{H_{2}{\mathrm{SiO}}_{3}}\right)$$

Rearranging:(5)$$\log\left(a_{H_{2}{\mathrm{SiO}}_{3}}\right)=\log\left(a_{{\mathrm{CaSiO}}_{3}}\right)-\log\left(a_{{\mathrm{Ca}}^{2+}}\right)-2\mathrm{pH}-\frac{\Delta G_{1}}{\mathrm{RT}}$$
(6)$$\log\left(a_{{\mathrm{HSiO}}_{3}{}^{-}}\right)=\log\left(a_{H_{2}{\mathrm{SiO}}_{3}}\right)+\mathrm{pH}-\frac{\Delta G_{2}}{\mathrm{RT}}$$

In the case of the reaction (R4), it can be seen from Equation (5) that the activity of H_2_SiO_3_, which is related to the amount of ion exchange, depends on the pH as well as the activity of Ca^2+^ ions in the solution. Moreover, the ion exchange will be prominent at a lower pH and lower activity of the Ca^2+^ ions in the solution. However, the activity of Ca^2+^ ions in the aqueous solution is large compared to Na^+^ ions. Therefore, the presence of Na^+^ ions in the glass leads to a higher ionic exchange mechanism. For reaction (R5), it can be seen from Equation (6) that the matrix dissolution only depends on the pH of the solution. Moreover, the matrix dissolution is larger at higher pH values, thus resulting in a greater dissolution of the matrix when the pH is >9 in experimental compositions.

The thermodynamic approach of Paul and Newton [147] suggests that glasses are a mixture of different silicate units. In other words, these silicate units would be *Q^n(ijkl)^* as seen by NMR spectroscopy and predicted by different network structure models. Therefore, the overall free energy of hydration (ΔG) can be written as sum of the individual contributions of the free energies of the hydration from the silicate units, given by:(7)$$\Delta G=\underset{i}{\sum }x_{i}\;\Delta G_{i}$$

Here, *x_i_* is the fraction of the component *i* in the glass composition. The silicate units with unknown ΔG_i_ are treated as oxides. The thermodynamic data on the formation of different components of the glasses could be obtained from the thermochemical tables [148,149]. 

The dissolution rate of a specific ion (*r_i_*) is defined as the amount of that ion released from a unit area per unit of time. Experimentally, the dissolution behaviour is characterised by measuring the change in the concentration of specific ions in the solution with respect to time and/or by measuring the weight loss of the sample material from a unit area with respect to time. However, the former gives a good description of the dissolution behaviour. Both of the quantities are related to each other by the quantity *V_S_*/*S_a_*, where *V_S_* is the volume of the solution, and *S_a_* is the surface area of the sample.
(8)$$r_{i}=\left(\frac{V_{S}}{S_{a}}\right)\frac{dC_{i}}{dt}$$

Here, *C_i_* is the concentration of the *i*th component, and *t* is the time. Moreover, the factor *V_S_*/*S_a_* also provides a means to compare experimental results with different *V_S_*/*S_a_* ratios. The ISO standard for SBF tests on the bioactive glasses [150] recommends a *V_S_*/*S_a_* ratio of 100 mm. However, from a practical experimental point of view, this is not convenient. Therefore, in a recent paper [151], we have suggested a *V_S_*/*S_a_* ratio of 20 mm with *V_S_* = 6 mL and *S_a_* = 3 cm^2^. Theoretically, the *V_S_*/*S_a_* ratio would not affect *r_i_*. However, the quantity *S_a_* is very difficult to measure, due to the fractal nature of the surfaces and time dependence of *S_a_*(*t*) during the experiment. Therefore, it seems necessary to standardise the quantities *V_S_* and *S_a_*. From a thermodynamics point of view, a closed system is a system where the overall components/mass is fixed. Whereas in an open system, there will be either (1) a transfer of the components/mass or (2) the pH of the solution is buffered. The ISO (International Organization for Standardization) standard for SBF tests on the bioactive glasses [150] recommends a closed system. However, some conventional laboratory practices involve periodic changing of the SBF, thus making the test system an open system. Researchers also employ Tris-buffered solutions, which would be an open system, to study the bioactivity and chemical degradation.

### 3.4. The Effects of Adding Other Components to the Na_2_O–CaO–SiO_2_ Glass System

Most of the bioactive glass compositions that have been developed so far typically belong to the ternary Na_2_O–CaO–SiO_2_ system, which is essentially the same compositional system adopted for common glasses such as windows, food and beverage containers, decorative tableware, etc. However, the bioactive glass compositions are enriched in network modifiers to make them less chemically durable [23]. The silica contents in bioactive glasses are usually less than 60 mol %, while Na_2_O and CaO content are relatively high. P_2_O_5_ is also often added as a fourth minor component, with the resulting compositions having a high CaO/P_2_O_5_ ratio [57,152]. Other constituents such as Al_2_O_3_, B_2_O_3_, MgO, or CaF_2_ might be also added with specific purposes. The bioactivity of a glass largely depends on its composition and surface reactivity. It is known that small variations in the glass composition strongly affect the final properties of bioactive glasses, such as the degradation rate and bioactive potential [23,153,154,155,156]. The *Class A* bioactive glass compositions usually lie in the ranges of 40%–52% SiO_2_, 10%–50% CaO, 10%–35% Na_2_O, 2%–8% P_2_O_5_, 0%–25% CaF_2_, and 0–10% B_2_O_3_. Glasses containing 45–52 wt.% of SiO_2_ featured the fastest rates of bonding, being able to bond to soft and hard connective tissue within five to 10 days. Increasing the content of SiO_2_ results in a decrease of bioactivity: glasses containing 55–60 wt.% SiO_2_ require a longer time to bond with bone, but did not bond to soft tissue, while glasses with >60 wt.% SiO_2_ are expected to be biologically inert [157,158]. 

All of the components of glass composition play specific roles, and their contents have significant effects on glass-forming ability, glass structure, and final properties [4,5,23,159]. Na_2_O acts as an effective fluxing agent, lowering the glass melting temperature, and *T_g_* and increases the dissolution from the glass surface, promoting the formation of a silica-rich layer that is necessary to the deposition of Ca^2+^ and P^5+^ ionic species that lead to the crystallisation of the bonding apatite layer. CaO and Na_2_O can be replaced by MgO and K_2_O, respectively, with little effect on bone bonding. The presence of MgO promotes the formation of apatite, and leads to the formation of an Mg-rich calcium phosphate layer. Mg also helps control the dissolution of the apatite precipitates. The partial substitution of CaO by CaF_2_ does not change the bone-bonding behaviour, but the presence of F leads to a decrease in the dissolution rate. P_2_O_5_ has an important role in bioactivity. For many years, it was assumed that the presence of P_2_O_5_ in glass compositions was crucial for bioactivity. However, it is now known that some phosphate-free glasses are bioactive. Al_2_O_3_ is important in controlling glass surface durability, as well as melting and forming characteristics, but in contrast to B_2_O_3_, it can inhibit bone bonding. Al_2_O_3_ can be incorporated in the glass composition, but the maximum Al_2_O_3_ amount depends on glass composition, being generally in the order of 1.0–1.5 wt.%, since glasses with more than 1.5 wt.% of Al_2_O_3_ lose their bioactivity [10]. Both Al_2_O_3_ and B_2_O_3_ have been used in bioactive glasses to change the surface reaction of glasses or the processing parameters.

#### Ion-Doped Bioactive Glasses

The physical and functional properties and the in vitro and in vivo performances of bioactive glasses can be modified and improved with the incorporation of doping oxides in trace amounts (e.g., Cu, Sr, Ag, Co, Zn, F, etc.) [23]. The incorporation of low concentrations of different ions is important to produce new functional materials with specific biological responses regarding the osteogenesis, angiogenesis, or antibacterial properties [160,161]. These ions allow obtaining beneficial therapeutic effects or permit functionalising the surface with biomolecules for additional effects to the tissue healing or regeneration process. It is known that the release of ions from the glass into the surrounding environment can activate various processes, leading to the growth of new tissues [65]. Besides the common ions that are present in the bioactive glass compositions that exist in the human body and play important roles in its biological activity, such as Si^4+^ and P^5+^ [162,163], many inorganic ions that are used as dopants (e.g., Mg^2+^, Sr^2+^, Zn^2+^, etc.) [164,165,166] are also present in human body, and have relevant effects in bone metabolism. Below are listed examples pointing out the biological role and the effect of some ions on glass bioactivity: (1)Strontium is an essential trace element of human body that has a beneficial effect on bone metabolism (promoting bone formation and osteoblast replication), enhances osteogenic differentiation, and helps stabilise the bone structure [167,168]. Bioactive glasses doped with strontium are able to increase the rate of a bone-like apatite layer formation on their surface, and show a fast decreasing of the Ca/P ratio, resulting in the stability of the apatite layer [169].(2)Magnesium is the fourth most abundant cation in the human body, being present in the natural enamel, dentin, and bone [170]; it plays an important role in bone metabolism [171]. The addition of magnesium to glasses in the SiO_2_–CaO–Na_2_O–P_2_O_5_ system influences the formation and the evolution of the newly formed layers, promoting the dissolution of the silica network, increasing the thickness of the silica gel layer formed conventionally prior to the apatite-like layer, and decreasing the crystallisation rate of the apatite layer [172]. Several in vitro studies revealed that Mg-doping favours cell adhesion, proliferation, and the differentiation of osteoblasts cells in comparison to control samples [173,174,175].(3)Zinc can be found in all of the biological tissues, being an essential microelement with an important role in bone metabolism. It induces bone formation in vitro, and prevents bone resorption [176]. Additionally, zinc has antibacterial and anti-inflammatory properties [177]. Zinc deficiency slows skeletal growth and causes alterations in bone calcification [178]. The bioactivity and biocompatibility properties of Zn-doped bioactive glasses are not only related to the apatite-forming ability, but also to the release of zinc ions that stimulate the proliferation and differentiation of bone-forming cells. The addition of zinc to alkali-free BG was observed to induce apatite formation. Although it might delay the nucleation of HCA at the early stage of SBF soaking, the HCA formation in long-term immersion is not affected.(4)Tantalum oxide was added up to 0.5 mol % to the SiO_2_–ZnO–CaO–SrO–P_2_O_5_ glass system by Towler et al. [179] at the expense of ZnO. Stronger bonds within the glass network without any adverse effects on the solubility of the glasses are reported the authors. Further, the Ta_2_O_5_ incorporation also resulted in extended working times and enhanced radiopacity, ion solubility, and long-term mechanical stability [180]. Furthermore, Ta_2_O_5_-containing glasses were also reported to possess antibacterial and antifungal activity against both Gram-negative (*Escherichia coli*) and Gram-positive prokaryotes (*Staphylococcus aureus* and *Streptococcus epidermidis*), as well as eukaryotes (*Fusarium solani*).(5)Copper is an essential micronutrient that is involved in many metabolic processes, including angiogenesis [181,182]. It plays a crucial role in bone formation and healing, and presents antibacterial properties. Insufficient amounts of copper can cause a reduction of bone mineral density [183]. Cu-doped BG showed antibacterial activity in suppressing some of the bacterial pathogens involved in postsurgical infections [184].(6)Silver shows antimicrobial activity, which makes it attractive for diverse applications such as surgical instruments, contraceptive devices, wound dressings, laundry detergents, wall paints, underwear, etc. Bioactive glasses doped with small amounts of silver ions showed a broad spectrum of antimicrobial activity, which can prevent infections [185,186]. For instance, the incorporation of 3 wt.% of Ag_2_O to the composition of a bioactive glasses in the system SiO_2_–CaO–P_2_O_5_ conferred antimicrobial properties to the glass without compromising its bioactivity [187]. Although the antimicrobial properties of silver can help treat or prevent infections, high Ag concentrations have been reported to be cytotoxic [188].(7)Cobalt is an essential element in human physiology and an integral part of B12, which is a vitamin that the human body is unable to produce. Bioglasses doped with cobalt are bioactive, develop a HCA layer on the surfaces in SBF, and show improved angiogenesis once implanted in bone [189,190,191].(8)Fluoride is a promising doping agent to enhance the biocompatibility of bioactive glasses, in particular in dental applications, because it provides a high acidic resistance of tooth enamel by substituting OH^−^ sites in dental apatite, leading to a partial conversion into fluorapatite [192,193], which features a much higher physicochemical stability, such as an increased resistance to dissolution by acid, than HCA [194]. Further, it prevents the demineralisation of enamel and dentin, improving remineralisation, and inhibiting bacterial enzymes.(9)Aluminium is the most abundant metal and the third most abundant element in the Earth’s crust. Due to its superior load-bearing capability and biostability, Al_2_O_3_ has been used in artificial (hip and knee) joints [195]. The addition of Al_2_O_3_ to bioactive glass improves the long-term stability and chemical durability, but hinders bioactivity [196]. The presence of Al in the bioactive glass composition can lead to the inhibition of bone bonding (due to decreasing the HCA formation rate), but up to about 1.5 wt.% Al_2_O_3_ can be incorporated into glass composition without significant negative impacts on the bioactivity [10].(10)The effects of adding TiO_2_ on the antibacterial behaviour, solubility, and cytotoxicity of silica-based and borate-based glass were also studied [197]. The TiO_2_-containing borate-based glasses exhibited significantly superior antibacterial and solubility behaviour in comparison to the silica-based glasses. The effects of partial replacements of SiO_2_ in silica-based glasses, or B_2_O_3_ in borate-based glasses, by TiO_2_ were investigated aiming at adjusting the coefficient of thermal expansion (CTE) of the glass coatings to that of Ti6Al4V substrate [198]. A better fit of CTE values was obtained for borate-based glass. This is important to prevent thermal stresses between the glass coatings and this alloy when used in medical implants, which is common.(11)Gallium oxide has been explored as an additive to zinc borate bioactive glasses to confer them with antibacterial properties [199]. The same authors also investigated the suitability of gallium-releasing zinc borate bioactive glasses for osteosarcoma-related bone graft operations using MTT and live/dead assay [200]. They concluded that glass powders containing 5 wt.% of Ga_2_O_3_ could enhance the viability of preosteoblasts while reducing that of osteosarcoma cells.

### 3.5. The Need for New Smart Approaches and Non-Biased Literature Surveys

Numerous books, book chapters, and papers have been written about bioactive ceramics, glasses, and glass-ceramics as reviewed above, especially in the Section “*3.1 Bioactive glasses in numbers*”. Most of the BG compositions that have been studied until the beginning of the 21st century contain high alkali percentages [158]. Decreasing the melting temperature was an obvious motivation behind [65]. Another reason was to increase the degradation of the silicate network over time [201,202]. This explains why alkali-free BG compositions used to be prepared preferentially by the sol-gel method [203]. The actual great interest on this subject is reflected in the important and continuous research activities in this field, covering broad ranges of chemical compositions and their reflexes on the glass structure, thermal properties, chemical durability/degradation, processing ability, and performances in vitro and in vivo [101,118,120,121,123,124,125,126,127,128,129,144,201,204,205,206,207,208,209,210,211,212,213,214,215,216,217,218,219,220,221,222,223,224,225,226,227,228,229,230,231,232,233,234,235,236,237,238,239,240,241,242,243,244,245,246,247,248,249,250,251,252,253,254,255,256,257,258,259,260,261,262,263,264,265,266,267,268,269,270,271,272]. However, only a few review papers that briefly refer to this kind of BG composition prepared by melt quenching [66,123,214] were published in the field starting from 2011 (three, about 13%), which was the year when most of the alkali-free BG contributions started to be added to the literature. However, the term ‘alkali-free’ could not be found in the ‘title’, ‘abstract’, or ‘keyword’ fields. It was necessary to dig deeper and include all the fields (‘any field’) in order to find any reference to alkali-free. As a matter of fact, most of the review papers published in the period from 2011 up to the present date (20, about 87%) [65,101,118,120,121,122,124,126,127,204,215,216,217,226,227,229,231,233,273,274] provide only a partial coverage of the broad topic, overlooking relevant contributions outside the mainstream. Moreover, the review papers cited above tend to deliberately omit the several disadvantages of high alkali-containing BGs [67,68,69,70,72,73,76,77,275]. Such high alkali content bioactive glasses are usually also hygroscopic [276], which is a serious drawback for applications in bioactive glass/polymer composites, affecting the stability, degradation, and mechanical performance of the composite materials. The presence of [OH^−^] ions on the surface of the glass powders promotes crystallisation. The leaching of high alkali contents induces in vitro cytotoxicity effects in cell culture media and in the living tissues around the implant due to the high local pH environment [67,76,77,275,277]. Such high pH environment favours the formation of HCA, but is likely to give false positive bioactivity results in SBF, while being unfavourable for homeostasis [76]. Excessive changes in the medium pH can inhibit osteoblast activity and cause cell necrosis or apoptosis [76,278]. Therefore, the pertinence of the continuous research activities focused on high alkali-containing bioactive glasses is highly questionable. As reviewed above, such glass compositions hardly can meet the most salient features of an ideal bioactive glass, concerning not only the in vitro and in vivo performances, but also the thermal, physicochemical properties, and processing ability, which include [279]:(1)Absence of cytotoxic effects (no harmful dissolution products and the resulting pH);(2)Non-genotoxic—no damage to genes within a cell or DNA mutations;(3)Biocompatible—absence of any foreign body reaction;(4)Fast biomineralisation rate in vitro with the formation of a hydroxyl carbonated apatite (HCA);(5)Osteoconductivity—bone readily grows and bonds on its surface;(6)Osteoinductive properties—recruiting immature cells and stimulating them to develop into pre-osteoblasts, which are essential in any bone healing process;(7)Osseointegration—stable anchorage of an implant achieved by direct bone-to-implant contact.(8)For implant coatings, good matching of the coefficients of thermal expansion of the coating glass and the metallic substrate for a strong adhesion between applied films and metallic implants;(9)Ease of scaffold fabrication by additive manufacturing techniques;(10)Ability to release therapeutic and anti-infection ions at the implant site.

Solving this complex challenge for multifunctional bioactive glasses with well-balanced properties requires new and smart approaches, which have been pursued by Ferreira et al. in several works published especially over the last decade [27,77,135,275,277,280,281,282,283,284,285,286,287,288,289,290,291,292,293,294,295,296,297,298,299,300,301,302]. Significant improvements in the overall properties were achieved using bioactive glass and glass-ceramic compositions with low alkali contents in the SiO_2_–Al_2_O_3_–B_2_O_3_–MgO–CaO–Na_2_O–F system [27,286,287,289,290,291,292,294,302]. These materials exhibited good sintering ability and excellent performances in vitro [27,286,287,289,290,302] and in vivo [291]. They were also used in the formulation of injectable devices [292,294]. However, when sodium oxide was gradually added to partially replace MgO in a series of glasses prepared by the melt quenching technique with compositions expressed as (16.20−*x*)MgO−*x*Na_2_O–37.14CaO–3.62P_2_O_5_–42.46SiO_2_–0.58CaF_2_ (in mol %), where *x* varied between zero and 10, it was observed that increasing sodium contents at the expenses of MgO induced: (i) a slight depolymerisation trend in the silicate glass network; (ii) an enhanced affinity of alkali cations towards phosphate; (iii) a delayed formation of the amorphous calcium phosphate surface layer upon immersing in SBF solution; and (iv) an increased cytotoxicity effect using mouse-derived pre-osteoblastic MC3T3-E1 cell line. These results suggested that alkali-free compositions could be a better bet to explore. The attempts made in this direction will be reviewed in the next section. 

### 3.6. Alkali-Free Bioactive Glasses

The most salient features desired for bioactive glasses as listed above can be obtained while totally excluding the alkalis and by a rational combination of all the remaining pertinent glass components, as has been plenteously demonstrated by Ferreira et al. in several works published since 2011 [77,135,151,275,277,279,280,281,282,283,284,285,298,299,300,303,304,305,306,307,308,309,310,311,312,313]. Following a completely different concept, the alkali-free bioactive glass compositions were based upon the compositions of minerals that are biocompatible and bioactive, such as diopside, fluorapatite, wollastonite, and tricalcium phosphate, in different combinations and proportions. The emphasis in this section is particularly put on alkali-free bioactive glasses and glass-ceramics as a smart way to overcome all the drawbacks mentioned above for high alkali containing compositions, as summarised elsewhere [279].

A series of bioactive glass compositions within the fluorapatite [FA; Ca_5_(PO_4_)_3_F]–diopside (Di; CaMgSi_2_O_6_) joined with varying FA/Di ratios (Table 9) were firstly synthesised by the melt quenching technique and investigated for their structure, apatite-forming ability, and physicochemical degradation [300,302]. Amorphous glasses could be obtained only for compositions up to 40 wt.% of FA. Silicon was predominantly present as a *Q*^2^ (Si) species, while phosphorus was found in an orthophosphate-type environment in all the investigated glasses. Furthermore, all of the glasses exhibited weight gains (instead of weight losses) upon immersion in citric acid buffer, which was used to study their physicochemical degradation in accordance with ISO 10993-14 ‘‘Biological evaluation of medical devices—Part 14: Identification and quantification of degradation products from ceramics’’. The in vitro cellular responses to glass-ceramics showed good cell viability and the significant stimulation of osteoblastic differentiation, suggesting the possible use of the glass-ceramics for bone regeneration [300]. Moreover, the sintering ability and the apatite-forming ability of glasses/glass-ceramics with FA contents within (10–25 wt.%) were significantly enhanced.

A particular composition of this system (80Di–20FA) was selected, and glass powders with different particle size distributions and mean particle sizes varying between 14–220 μm were used for detailed sintering studies [280]. An array of complementary advanced characterisation techniques including in situ high-temperature scanning electron microscopy (HT-SEM), differential thermal analysis (DTA), hot stage microscopy (HSM), X-ray diffraction (XRD), and scanning electron microscopy (SEM) was adopted to gather relevant data. It was shown that irrespective of the mean particle size, the bioglass exhibited good sintering ability. 

Neck formation and other morphological changes observed by HT-SEM were initially driven by surface diffusion at temperatures apparently below *T_g_* but without noticeable macroscopic shrinkage. With temperatures increasing above *T_g_*, the particles formed individual spherical droplets, which then merged into larger liquid droplets, which were signs of their excellent sintering ability. Further increasing the temperature up to about 850 °C led to incipient formation diopside and fluorapatite crystals embedded into a high (~90 wt.%) residual glassy phase matrix. The content of residual glassy phase tended to increase as the mean particle size increased.

Based on the most interesting results gathered in the frame of previous works [280,300,302], another series of bioactive glass compositions (Table 10) were designed within the diopside–fluorapatite–wollastonite (W-CaSiO_3_) ternary system [CaMgSi_2_O_6_]80−*x*[Ca_5_(PO_4_)_3_F]20–[CaSiO_3_]*x* (*x* = 10–80 wt.%), starting from the parent glass composition (Di80–20FA) [275]. The aim of adding W was to further improve the sintering ability that was investigated by differential thermal analysis (DTA). The wider sintering window (~145 °C) was observed within the range of W-10–W-30, followed by a gradual narrowing trend with further increasing W contents. It was concluded that varying the CaO/MgO ratio on glasses did not exert any significant effect on the structure of glasses, with Si predominantly present in *Q*^2^ units, while phosphate is coordinated in an orthophosphate environment. With respect to the thermal behaviour of glasses, heat-treating glass powder compacts at 850 °C for one hour resulted in well-sintered glass-ceramics with diopside, fluorapatite, wollastonite, and pseudowollastonite as the crystalline phases. Increasing the CaO/MgO ratio in glasses degraded their sintering behaviour and resulted in different amorphous/crystalline ratios in the resultant glass-ceramics. The glass-ceramics W-10–W-30 exhibited the higher amounts of residual glassy phase that favoured bioactivity.

New series of alkali-free bioactive glass compositions were designed in the diopside (Di; CaMgSi_2_O_6_), fluorapatite (FA; Ca_5_(PO_4_)_3_F), and tricalcium phosphate (TCP–3CaO.P_2_O_5_) system, combined in different proportions [275,284,300,301,314]. Figure 5 shows some investigated compositions in the ternary Di–FA–TCP diagram, as well as in the binary Di–FA [280,300,302] and Di–TCP systems.

Table 11 provides the compositional details of the most interesting BGs in this ternary system. The glass-forming ability and stability of these glasses strongly depended on the composition and the cooling rate. Fluorapatite-richer compositions were less prone to glass formation and underwent fast crystallisation, even upon quenching the melts in cold water to obtain the glass frits (black symbols). Among the diopside-richer compositions, some enabled obtaining amorphous frits, but the bulk glasses cast on metal plates tended to partially crystallise, especially in the parts further from the metal plates that cooled more slowly (white core symbols), while others enabled obtaining amorphous materials (grey symbols). These last ones were the most interesting compositions from the processing viewpoint. The silicate network consisted predominantly of *Q*^2^ (Si) units, while phosphorus tends to remain in an orthophosphate (*Q*^0^) environment, which is a common feature to all the designed alkali-free formulations. Some of the investigated glasses exhibit HCA formation on their surface within one to 12 hours of their immersion in SBF solution [275]. The composition 70Di–10FA–20TCP exhibited a particularly fast biomineralisation capability with the formation of a crystalline surface HCA layer after immersion in SBF solution for one hour. 

Since the bonding to living tissues after implantation is mediated by this HCA layer, a fast bonding capacity is expected from these bioactive glass compositions, and especially from the 70Di–10FA–20TCP (TCP-20) one. Due to this, this composition was also designated as FastOs^®^BG. The alkaline phosphatase activity and osteogenic differentiation using rat bone marrow mesenchymal stem cells seeded on sintered glass powder compacts revealed that the tested compositions are ideal potential candidates for applications in bone tissue engineering. 

FastOs^®^BG glass demonstrated osteogenic activity, inducing the differentiation of human mesenchymal stem cells (hMSCs) into bone-forming cells, even in the absence of osteogenic medium [284]. This osteoinduction effect was significantly greater in comparison to that of 45S5 Bioglass^®^. The in vivo performance of FastOs^®^BG was tested in a sheep animal model and compared with that of 45S5 Bioglass^®^ as control [306]. Histological and scanning electron microscopy assessments of retrieved subcutaneous and bone samples demonstrated that FastOs^®^BG is more slowly resorbed, more biocompatible and osteoconductive, and more easily osteointegrated in comparison to 45S5 Bioglass^®^. Therefore, FastOs^®^BG has greater potential as a bone graft material for large bone defects. 

Moreover, the sintering ability, as investigated by differential thermal analysis and hot-stage microscopy, revealed that fully dense amorphous materials could be obtained upon sintering at 800 °C. These features are essential for the fabrication of mechanically strong bioactive glass scaffolds for bone regeneration. 3D porous scaffolds with pore sizes of 200 μm, 300 μm, and 500 μm could be easily fabricated from FastOs^®^BG by additive manufacturing [285]. Printable inks containing 47 vol.% solids with rheological properties tuned to meet the stringent requirements of robocasting technique could be obtained. The fully densified filaments obtained upon sintering conferred to the scaffolds compressive strength values that were higher in comparison to cancellous bone. A similar BioExtrusion technique was used to produce polycaprolactone (PCL)–bioglass (FastOs^®^BG) composites scaffolds containing 20%, 30%, and 35% bioglass [305]. The addition of bioglass was found to decrease the elastic gradient and yield stress if two scaffolds of the same density are compared.

The ease of scaffold fabrication by robocasting from the alkali-free FastOs^®^BG composition contrasts with the cumbersome processing when starting from 45S5 Bioglass^®^ powders. The first successful deposition of 3D porous scaffolds from 45S5 Bioglass^®^ by Ferreira et al. [315,316] required a completely new approach to solve the dispersion problems and coagulate the suspensions. Such problems could only be overcome using carboxymethylcellulose (CMC) as a single multifunctional (dispersant, binder, gelation agent) processing additive, enabling obtaining aqueous suspensions with 45 vol.% solids. However, the typically poor sintering ability of 45S5 Bioglass^®^ led to insufficient densification, extensive crystallisation, and poor mechanical strength. 

The superiority of FastOs^®^BG justified its selection as the parent glass composition for another study aiming at investigating the influence of partially replacing CaO by SrO regarding the structure, apatite-forming ability, physicochemical degradation, and sintering behaviour of a new bioactive glass series with the composition (mol %): (36.07−*x*)CaO–*x*SrO–19.24MgO–5.61P_2_O_5_–38.49SiO_2_–0.59CaF_2_, where *x* varies between zero and 10 [299]. The results revealed that the Sr^2+^/Ca^2+^ ratio did not significantly affect the glass structure, but the apatite-forming ability of glass powders immersed in SBF for time durations varying between one hour and seven days decreased considerably. Further, the addition of strontium led to a sevenfold decrease in chemical degradation of glasses in Tris–HCl and citric acid buffer. Sintering glass powder compacts for one hour at 850 °C resulted in full dense GCs with a residual glassy phase between 31 wt.% and 47 wt.%. Di was crystallised as the dominant phase, and FA was also formed as the secondary phase. The flexural strength values of GCs varied between 98–131 MPa. The large amounts of residual glassy phase, along with the good flexural strength, proved the potential of the developed GCs for the scaffold fabrication in bone tissue engineering. 

The foremost studies on Sr-containing bioactive glasses were published in 1995 by Galliano et al. [317,318]. The authors were not yet sure about their biocompatibility. The subject did not attract further attention for more than one decade, but the interest in Sr-doped bioactive glasses was rejuvenated as deduced from a series of Sr-containing bioactive glass compositions patented by Hill and Stevens [319] and Jallot et al. [320], and from the boom in the number of scientific publications [169,321,322,323,324]. These studies demonstrated the benefits of Sr-doping bioactive glass for their in vitro performance. Recently, Hill and Stevens [319] patented a series of strontium-containing bioactive glass compositions, among which a glass with (wt.%) 44.08SiO_2_–24Na_2_O–21.60CaO–4.43SrO–5.88P_2_O_5_ is being commercialised as Stron-Bone™ by RepRegen Ltd (London, UK). 

A similar study was also carried out starting from FastOs^®^BG as the parent glass composition and partially replacing MgO by ZnO in the composition (mol %): 36.07CaO–(19.24−*x*)–MgO–*x*ZnO–5.61P_2_O_5_–38.49SiO_2_–0.59CaF_2_, where *x* varied between zero and 10 [281,298]. The aim was to investigate the effects of the Zn^2+^/Mg^2+^ ratio on the structure by molecular dynamics simulations and nuclear magnetic resonance spectroscopy. The network connectivity of these glasses is lower than that reported for 45S5 Bioglass^®^ [325]. An increase in the Zn^2+^/Mg^2+^ ratio did not induce any significant change in *Q^n^* speciation and network connectivity; however the chemical durability of the glasses was improved, and tended to suppress the carbonated-HA forming ability in SBF [298]. This was attributed to the fivefold M–O coordination of Mg^2+^ and Zn^2+^ and the glass network strengthening effect of Zn through non-covalent Si–O–Zn–O–Si links [298].

The other parallel work [281] aimed at investigating the influence of the partial replacement of MgO by ZnO on the structure, sintering ability, crystallisation behaviour, and bioactivity. The ZnO content was revealed to play an essential role in the in vitro bioactivity. The proliferation of mesenchymal stem cells (MSCs) and their alkaline phosphatase activity (ALP) on GCs revealed to be Zn-dose dependent with the highest performance being observed for 4 mol % ZnO, followed by a clear decreasing trend with further increasing ZnO contents [281].

The influence of SrO and ZnO co-doping on thermomechanical behaviour of alkali-free bioactive glass-ceramics in the system: (mol %) (36.07−*x*)CaO–*x*SrO–(19.24−*y*)MgO–*y*ZnO–5.61 P_2_O_5_–38.49SiO_2_–0.59CaF_2_ (*x* = 2–10, *y* = 2–10) was investigated in another work [283]. The detailed compositions are reported in Table 12.

Hot-stage microscopy and differential thermal analysis revealed that the densification of all the glass powders occurs before the onset of crystallisation, resulting in fully densified and mechanically strong glasses/glass-ceramics after heat treating for one hour at 800 °C, 850 °C, and 900 °C. The crystalline phase assemblage for sintering temperatures >800 °C included diopside (CaMgSi_2_O_6_) and fluorapatite (Ca_5_(PO_4_)_3_F). The results obtained revealed that a better balance of properties was achieved for the samples sintered at 850 °C, providing insightful criteria for selecting the material/experimental conditions for the development of three-dimensional porous scaffolds for bone tissue engineering.

The same compositions that were reported above in Table 12 were further investigated for the structure–property relationships in order to shed light on the structural role of co-doping ions (Sr^2+^ and Zn^2+^) on the chemical dissolution behaviour of glasses and its impact on their in vitro bioactivity [135]. The relevant structural properties, which were assessed by molecular dynamics simulations in combination with solid-state NMR spectroscopy were well correlated to the degradation behaviour, in vitro bioactivity, osteoblast proliferation, and alleviation of the oxidative stress levels exerted on the human osteosarcoma MG63 cell line. A dose-dependent cytoprotective effect of glasses with respect to the concentrations of zinc and strontium released was observed, enhancing the cell viability and negating the effect of oxidative stress induced by the addition of H_2_O_2_ to the cell culture medium. This cytoprotective effect of co-doped glasses contracts well with the cytotoxicity effect exerted by 45S5 Bioglass^®^ [76,77,82,85,135].

Glass compositions in the binary system diopside (CaO·MgO·2SiO_2_)–tricalcium phosphate (3CaO·P_2_O_5_) with varying Di–TCP proportions (Table 13 below). 

The effects of non-isothermal heating for one hour at 900 °C, 1000 °C and 1200 °C on the structural changes, sintering ability, crystallisation behaviour, and three-point bending strength were investigated by hot-stage microscopy, differential thermal analysis, and magic angle spinning NMR and XRD techniques [282]. Amorphous glasses could only be obtained from compositions with Di ≥ 50 wt.%. Glasses with Di ≥ 60 wt.% exhibit sintering temperature windows that are wide enough for the fabrication of scaffolds. Diopside and hydroxyapatite were the major phases present at 900 °C and 1000 °C with calcium silicate and whitlockite as the minor phases. At 1200 °C, diopside and whitlockite were the major phases present, with no other minor phase detected [282].

The most promising bioactive glass compositions (Di-60 to Di-90) reported in Table 13 were further investigated for the in vitro performance using human mesenchymal stem cells (hMSCs). Significant statistical increases in the metabolic activity of hMSCs when compared to the control were observed for Di-60 and Di-70 glasses under both basal and osteogenic conditions [284]. 

All of the investigated glasses underwent considerably lower weight losses in Tris–HCl in comparison to that of 45S5 Bioglass, but exhibited enhanced in vitro biomineralisation activity expressed by the formation of an HCA surface layer over seven days of soaking in SBF. 

An insightful assessment of the biological performances offered by several alkali-free bioactive glasses reported above was made using 45S5 Bioglass^®^ as control [277], as reported in Table 14 below. The aim was to evaluate and compare their abilities to stimulate human mesenchymal stem cells (hMSCs) differentiation into osteoblasts.

The von Kossa assay demonstrated that all the bioactive glasses that were studied were able to induce the appearance of calcium deposits, even when the cells were cultured in Dulbecco’s modified Eagle’s medium (DMEM), proving that these biomaterials per se induce hMSCs’ cell differentiation. It was also observed that in both of the cell culture media tested (Dulbecco’s modified Eagle’s medium, and osteogenesis differentiation medium), alkali-free bioactive glasses clearly induced the appearance of more calcium deposits than 45S5 Bioglass^®^, indicating their greater ability to induce cell differentiation. These results clearly demonstrated the superiority of alkali-free bioactive glasses in stimulating the differentiation of hMSCs into bone-forming cells, making them a safe and better alternative materials for dental, orthopaedic, and maxillofacial surgery applications in comparison to 45S5 Bioglass^®^ [277]. 

### 3.7. Sputtered Bioglass Thin Films: A Reliable Biofunctionalization Option for Implantology

Nowadays, the international market for osseous implants and endoprostheses is dominated by the medical devices fabricated from titanium (Ti) or its medical-grade superalloys. Respectable financial analysis agencies forecast an outstanding growth of the implants market, in the response to the increasing health and societal needs [311,326,327]. The dental implants market segment alone, which was valued in 2016 at ~$3.77 billion [326], is projected by 2024 to exceed $5.2 billion, according to Global Market Insights, Inc. (Cleveland, OH, USA) [327] and $6.5 billion according to MarketWatch (San Francisco, CA, USA) [328]. In spite of their proven biocompatibility and excellent mechanical performance, the metallic implantable devices entail long healing periods, as they lack innate bioactive properties, and thereby do not possess the ability to induce a fast bonding with bone tissue. Currently, these limitations are attempted to be surpassed by applying bioactive coatings (with designed physicochemical features and biological performances) on the surface of the Ti or Ti-based implants and/or endoprostheses [329,330]. 

The only commercially available solution is the coating of metallic features with thick (>50 µm) layers of HA, an osteoconductive material, by the plasma spray method. No commercially (ready for clinical use) bioactive glass implant coatings yet exist, to the best of our knowledge. The benefits that could emerge from the delineation of an implant coating design based on bioactive compounds that are superior to HA (i.e., bioactive glasses with a higher index of bioactivity), explains the intensive research performed worldwide in the last period [211,331,332,333]. Consequently, a wide palette of deposition methods [334] has been explored over the years to achieve this conceptual desiderate (i.e., a mechanical and biological reliable bioactive glass implant coating layer), amongst which the most prominent fabrication techniques are: (i) enamelling/glazing [335,336,337,338], thermal spray [339,340,341,342], and electrophoretic deposition [343,344,345,346] for thick coatings (>5 µm—hundreds of µm), and (ii) sol-gel [347,348,349,350], pulsed laser deposition [351,352,353,354,355], and ion-beam [356,357] and radio-frequency magnetron [151,358,359,360,361,362,363,364]) sputtering for thin films (<5 µm). 

If the progresses achieved by enamelling/glazing, thermal spray, electrophoretic, sol-gel, and pulsed laser deposition techniques have been extensively debated in quite a large number of overview/summary articles/book chapters [211,331,332,333,334,365,366], the advances accomplished by the sputtering methods have been either completely omitted [211,331,334,366], or in the best case, just sparsely mentioned as a possible variant [332,333,365], using for instance an outdated early study [333]. Thereby, in this short section, the authors aim to complete the whole picture, and briefly present the results achieved in the realm of bioactive silica-based glass (SBG) thin films synthesised by ion-beam and radio-frequency magnetron sputtering, offering a flavour of their future potential.

The sputtering method is a prominent member of the physical vapour deposition (PVD) family, and is currently expansively used in the semiconductor and decorative industries. Its simple process relies on the expulsion of atoms from a target by bombardment with energetic ions (typically argon). The positively charged gas ions are attracted to the negative biased cathode target at very high speeds, resulting in the ejection of atoms, which are deposited in form of a film on a substrate conveniently positioned in the vicinity of the target. Any type of material can effectively be sputtered on virtually any type of solid substrate, since sputtering is typically a low-temperature process (<100 °C).

The ion-beam variant possesses the advantage that some of the deposition variables (i.e., angle of incidence, flux, and energy of ions) can be precisely and independently controlled, allowing for a more accurate engineering of the deposited film stoichiometry. Furthermore, the absence of plasma between the substrate and the target (present in the case of magnetron sputtering) permits the deposition of films onto materials that are highly sensitive to temperature. However, ion-beam sputtering yields reduced deposition rates with good uniformity only on constricted substrate areas, which does not make it very attractive for large-scale applications.

To date, only two attempts to prepare SBG glass films by ion-beam sputtering have been recorded. The first study was performed in 1982 by Larry Hench [356], the inventor of the first bioactive glass formulation (45S5, Bioglass^®^), as mentioned in the previous sections of the review. Films with thicknesses in the range of 0.5–4 µm, derived from the 45S5 Bioglass^®^ (mol %: SiO_2_–46.1, CaO–26.9, P_2_O_5_–2.6, Na_2_O–29.9) and 52S4.6 (mol %: SiO_2_–52.1, CaO–23.8, P_2_O_5_–2.6, Na_2_O–21.5) systems, have been deposited onto metallic (i.e., 316 L stainless steel), ceramic (i.e., alumina), and polymeric (i.e., poly(methyl metacrylate)) substrates. A good degree of substrate coverage was attained only for the thicker films. The most promising results have been achieved for the SBG-coated polymeric substrates, as indicated by the in vivo tests performed on animal model (Sprague–Dawley rats). The tissue adhesion of collagen to the SBG coating surface was observed. The second topical study followed 20 years later, with Wang et al. [357] reporting on the completely amorphous films synthesised onto Ti substrates from a low silica SBG formulation (mol %: SiO_2_–35, CaO–50, P_2_O_5_–15). The dense and homogenous coatings had excellent bonding strength (no delamination being observed by performing scratch tests at a load of 100 gf) and induced the proliferation of MC3T3-E1 mouse osteoblast cells.

In magnetron sputtering, the gun (cathode) design uses magnetic fields to trap the stray electrons in the vicinity of cathode target surface, which disallows these particles to bombard and heat (with possible damages) the growing film. Simultaneously, the ionisation probability of the neutral working gas molecules is augmented by several orders of magnitude, having as an effect a more stable plasma and greater number of available bombarding ions, and thus, a significant improvement of the sputtering yield [367]. Furthermore, radio-frequency magnetron sputtering (RF-MS) has a series of other remarkable advantages, including: high purity, excellent adherence, uniformity in both thickness and composition on large-area substrates (dependent only on the magnetron gun size), compactness, ability of facile engineering of film properties by variation of the process parameters (working pressure, electric power, target-to-substrate separation distance, substrate temperature, working gas composition), or the possibility of coating complex-shaped objects if adopting planetary rotation holders [367,368,369]. The demonstrated ability of RF-MS to surpass technological barriers and be scaled up with ease to an industrial level (as shown in the optoelectronics industry) [367,369] should increase its desirability in the biomedical field also. 

The first study assaying the deposition of bioactive SBG layers by RF-MS has been published only in 2003 by Mardare et al. [358]. Following this peripheral attempt, there can be identified in the main scientific data bases (e.g., Web of Science^®^, Scopus^®^, Pubmed^®^) significant research efforts devoted by the groups led by Jansen and van den Beucken [359,361,370,371,372], on SBG—hydroxyapatite mixed layers, and Stan and Ferreira [151,304,307,308,309,310,311,312,313,362,363,373,374], (on pure SBG thin films, with only isolated trials published by Slav et al. [360] and Saino et al. [364]. Recently, Stuart et al. [375,376,377,378,379] debuted a series of studies on the preparation by RF-MS of resorbable phosphate bioglass (PBG) thin films that are capable of releasing various therapeutic (e.g., antimicrobial) ions in a fast, but controllable manner.

From the plethora of existing SBG formulations [66,118], only a limited number of eight individual compositions (hereby given in mol %) have been employed so far in RF-MS studies: SiO_2_–31.1, CaO–31.7, P_2_O_5_–10.6, MgO–26.6 [304,358]; 45S5 [360,363,373,374,380]; SiO_2_–58.5, CaO–17.1, P_2_O_5_–4.5, MgO–7.9, Na_2_O–5.2, K_2_O–6.8 [362,380]; 1-b (SiO_2_–39.9, CaO–31.0, P_2_O_5_–2.6, MgO–13.3, Na_2_O–4.4, B_2_O_3_–4.4, CaF_2_–4.4) [307,308,309,313]; 58S (SiO_2_–59.8, CaO–38.5, P_2_O_5_–1.7) [364]; 1-d (SiO_2_–45.5, CaO–30.3, P_2_O_5_–2.6, MgO–13.0, Na_2_O–4.3, CaF_2_–4.3) [310,311,312]; S53P4 (SiO_2_–53.8, CaO–21.8, P_2_O_5_–1.7, Na_2_O–22.7) [359,361,370,371,372]; and Zn4 (SiO_2_–38.5, CaO–36.1, P_2_O_5_–5.6, MgO–15.2, ZnO–4.0, CaF_2_–0.6) [151].

Remarkable biomineralisation capabilities (i.e., formation of biomimetic apatite layers on the surface of the SBG RF-MS films) have been observed by performing in vitro assays in both (i) the purely inorganic simulated body fluid solution introduced by T. Kokubo in the early 1990s (now part of the ISO 23317:2014 bioactivity testing protocol) [151,359] and (ii) inorganic–organic media with increasing degrees of biomimicry, reproducing the human intercellular environment with higher fidelity [151]. SBG films have shown as well to have excellent cytocompatibility with various cell lines: rat bone marrow [361], SAOS-2 human osteoblast [364], primary human osteoblast [374], 3T3 mouse fibroblast [310], human dental pulp stem [311], HUVEC-Hs27 human umbilical vein endothelial [312], and human mesenchymal stem cells [151]. The ability of SBG RF-MS films to either (i) promote the proliferation and differentiation of osteoblast cells [364] or (ii) conserve an undifferentiated phenotype of stem cells, while enabling a good cellular adhesion and proliferation [311], has been reported.

Hitherto, in vivo tests on animal models (Beagle dogs’ mandibles [370] and Dutch Saane milk goats’ iliac crests [372]) have been performed only in the case of SBG–hydroxyapatite mixed RF-MS coatings derived from the S53P4 compositional system. In Ref. [370], it is mentioned that the films had a decreased silica content [from 52.7 wt.% to <40 wt.%]; thereby, one should expect high dissolution rates. Contradictory biological results have been obtained. The S53P4 addition to hydroxyapatite dental implant coatings resulted in inferior in vivo performances [370], while in the case of iliac implants coatings, the S53P4 incorporation into hydroxyapatite led to an enhanced in vivo biological behaviour [372], with respect to the pure hydroxyapatite control. However, no pure SBG control coating has been included in the experimental groups, due to the authors fearing the possible low adherence strength of the SBG films [370].

Indeed, one obstacle in the fabrication of mechanically reliable SBG coatings for implant applications was the significant mismatch between the coefficients of thermal expansion (CTE) of classical SBG systems (i.e., ~13–17 × 10^−6^ °C^−1^) and pure Ti and its medical-grade alloys (i.e., ~9.0–9.6 × 10^−6^ °C^−1^), which drastically limits the interfacial bonding strength. The decrease of SBG CTE can be accomplished by increasing the silica content (at the expenses of bioactivity) [362,380] or partially replacing (with MgO, B_2_O_3_, and/or CaF_2_ [290,309,310,311]) or even removing alkalis [151]. Furthermore, the high concentration of Na_2_O that is characteristic of 45S5 Bioglass^®^ and S53P4 BonAlive^®^ classical SBG systems, is known to induce faster degradation rates [279,380], which conflicts with the concept of durable thin implant coatings with long-lasting performance.

The fabrication of RF-MS coatings with good bonding strength (~50 MPa) from high CTE SBG systems (e.g., 45S5) can become possible if coupling (i) the insertion of a buffer layer with compositional gradient SBG_x_Ti_1−x_ (x = 0–1) at the Ti-SBG coating interface, with (ii) a post-deposition annealing treatment performed at a moderate temperature [373,374]. 

In the first report dealing with the fabrication of pure SBG layers by RF-MS, which is topical for this review, Mardare et al. [358] used for the cathode target disks obtained by the sintering at 1200 °C of glass powders with the following composition (mol %): SiO_2_–31.1, CaO–31.7, P_2_O_5_–10.6, MgO–26.6. A bonding strength (tested by the pull-out method) of ~41 MPa was obtained for the as-deposited amorphous low silica glass films onto Ti substrates. However, as a consequence of the inexplicable application of post-deposition thermal treatments at extreme temperatures (i.e., 900–1000 °C) situated well above the allotropic α→β phase transformation of Ti (which is accompanied by volume modification), and disregard for the large CTE mismatch between Ti and SBG, the films presented profound cracks. Thereby, their adherence to the substrates decreased drastically to ~16 MPa. This might have casted an undeserved shadow on the RF-MS capability to yield quality SBG layers. Ten years later, Stan et al. [304] showed that in order to achieve a good bonding strength of a film derived from the same SBG compositional system, the post-deposition heat treatments should be performed at temperatures slightly above the glass transformation temperature (i.e., 750 °C), such as to induce interdiffusion processes at the SBG/Ti interface. A pull-out adherence value of ~60 MPa has been obtained, which is higher than the one (i.e., 50.8 MPa) imposed by the most exigent international standard for implant coatings [381].

By using SBG systems with (i) high silica content [362] or (ii) moderate silica content and low alkali concentration [309,310], and thereby lower CTE values (i.e., 9.2–10.7 × 10^−6^ °C^−1^), excellent pull-out adherence values (i.e., (i) exceeding 85 MPa [362] or (ii) being situated in the 63–75 MPa range [309,310]) have been obtained for RF-MS films deposited under argon working pressures of 0.3–0.4 Pa. Promising values of hardness and elastic modulus, of ~5.7 GPa and ~77 GPa, respectively, have been achieved for SBG RF-MS films with a thickness of ~1 µm [310]. These values are similar to the ones obtained for plasma spray coatings with thicknesses of the tens or even hundreds of micrometres [339,342]. These progresses encouraged the application of such optimised RF-MS regimes for the coating with SBG layers of real dental implant fixtures [311]. The mechanical durability of the SBG RF-MS films has been tested by the “cold” implantation procedure in pig mandibular bone from a dead animal, followed by immediate tension-free extraction tests. No signs of SBG delamination from the surface of the Ti dental screws have been identified by the coupled SEM/electron dispersive X-ray spectroscopy multiple analyses, which advocated for their high bonding strength [311].

Noteworthy, it was recently shown that one can take advantage of the RF-MS film growth mechanisms (i.e., almost instantaneous condensation of vapour species, with low transverse mobility of the ad-atoms) to fabricate sub-micrometre hollow cone-shaped needles, which were envisioned as medical devices for focal transitory permeabilisation of the blood–brain barrier in the treatment of carcinoma and neurodegenerative disorders [312].

While RF-MS possesses a large number of advantages, it has also been criticised for its presumed inability to congruently reproduce with ease the composition of complex materials, including SBGs and PBGs, since the lighter species are more readily ejected from the target surface. However, by the variation of the main RF-MS deposition parameters (sputtering pressure, working atmosphere, target powder density, target-to-substrate separation distance), one can modify the composition and structure of films starting from a single target, in the pursuit of improved mechanical and biological performances [304,308,309,310,312,375,376,377,378,379]. Nevertheless, a longer pre-sputtering treatment of the target, performed prior to the film deposition process, is known to lead to the stabilisation of the sputtering processes and a more equilibrated heavier/lighter elements ratio at the target surface [312]. Thereby, although demanding, the replication of the target’s composition seems feasible, and desirable when taken into account the certain promises presented in this section of the review.

In conclusion, the RF-MS technique can be considered a genuine alternative for the biofunctionalisation of the next generation of osseous implants and endoprostheses. The path towards the manufacture of mechanically reliable SBG implant coatings has now been reopened by the recent delineation of SBG compositional systems, eliciting both low CTEs and unaltered biological properties. 

## 4. Concluding Remarks

The great importance of Hench’s pioneering work in discovering the bioactive glasses cannot be overemphasised. His discoveries inspired many other studies with different perspectives. Most of them are still focused on the original 45S5 Bioglass^®^ composition or on compositions based on it; it is probably one of the most often investigated materials. The pertinence in further exploring this perspective is quite questionable, as the degree of novelty is not clear, and it seems to go beyond science and towards advertisement. However, numerous available literature reports about bioactive glasses and glass ceramics provide plenty of evidence that high alkali-containing bioactive glasses present several main limitations, as have been detailed and reviewed in the sections above, including the abstract. 

Another completely different perspective is behind an increasing number of other studies that have aimed at tackling the specific problems posed or left unsolved by high alkali-containing bioactive glass compositions and emphasising the demonstrated advantages of using alkali-free or low alkali-containing bioactive glass compositions. From sections 3.5 and 3.6, it can easily be concluded that such alternative BG compositions can offer a set of well-balanced overall properties for the most demanding applications in healthcare, bone regeneration, and tissue engineering. Their 10 most important advantages are summarised in Section 3.5. Unfortunately, due mostly to artificial market barriers, the process by which new biomedical devices reach the market is cumbersome and discouraging, even when there is enough evidence about the comparative advantages offered by the new materials or devices in comparison to those being commercialised. The FastOs^®^BG is actually being pushed to the market by Ossmed-Regeneration Technology, S.A. Its superior overall properties suggest excellent promise for biomedical applications in dentistry, orthopaedics, maxillofacial surgeries, scaffolds fabrication for bone regeneration and tissue engineering, and as coating material for the surface functionalisation of metallic or ceramic implants.

## Figures and Tables

**Figure 1 materials-11-02530-f001:**
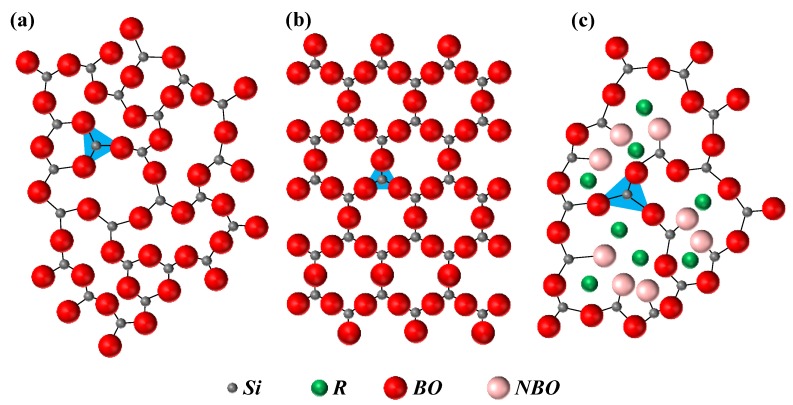
Two-dimensional representation of network structure of (**a**) vitreous and (**b**) crystalline silica; (**c**) vitreous silicate (*R* stands for a generic network modifier cation, while *BO* and *NBO* stand for bridging oxygen and non-bridging oxygen, respectively).

**Figure 2 materials-11-02530-f002:**
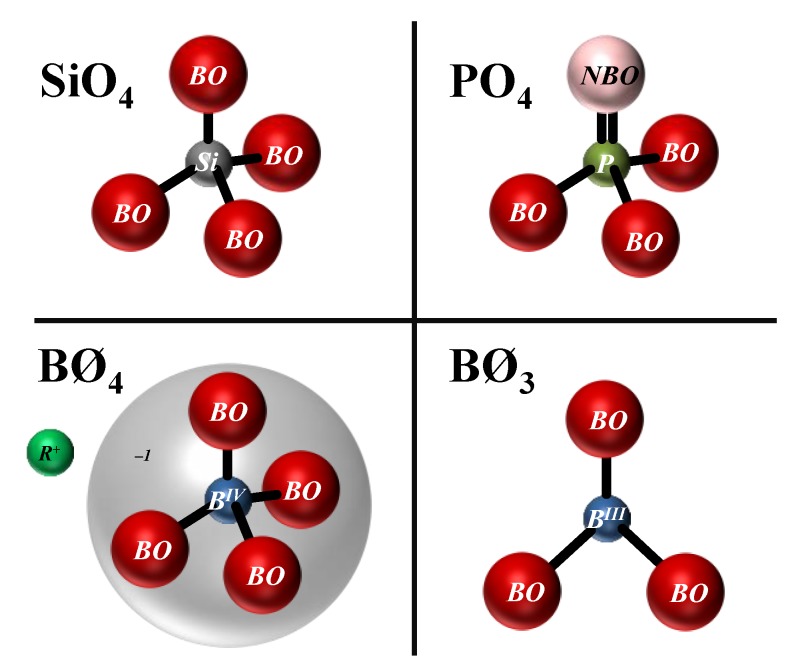
Schematic representation of structural units that build up an oxide glass network (*R* stands for a generic network modifier cation; Ø corresponding to *BO* atoms).

**Figure 3 materials-11-02530-f003:**
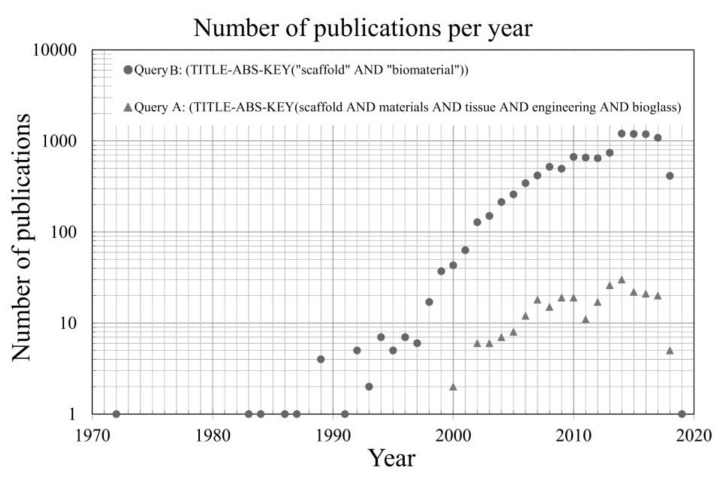
Number of publications of queries A and B in the timeline window between 1970–2020 (results obtained at the moment of the query search results); the Y-axis (number of publications) is presented in logarithmic scale.

**Figure 4 materials-11-02530-f004:**
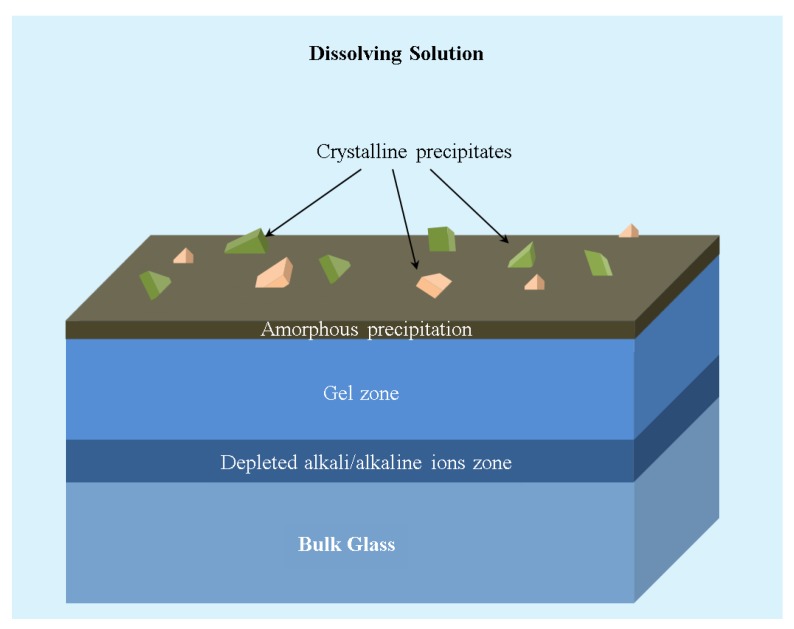
Schematic representation of the different zones.

**Figure 5 materials-11-02530-f005:**
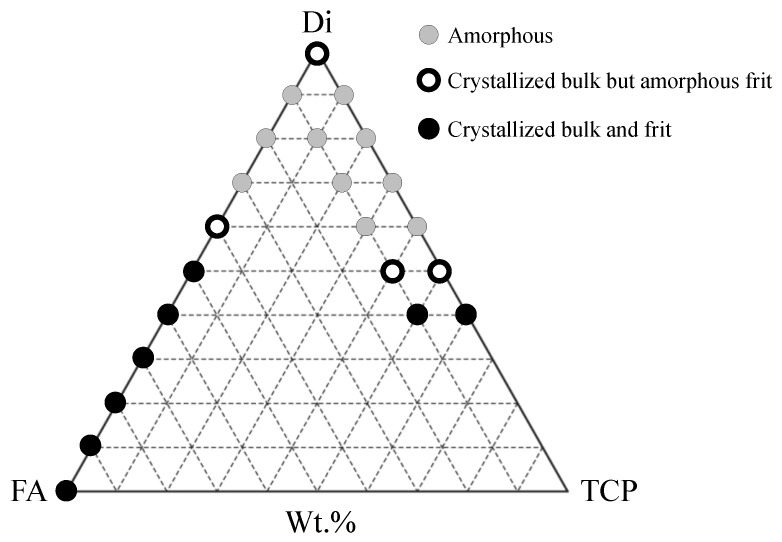
Graphical representation of the alkali-free bioactive glass compositions investigated [275,284,300,301,314].

**Table 1 materials-11-02530-t001:** Publications by access type.

Query	Access Type	Number of Items	Percentage
A	Open access	8	3.03%
	Other	256	96.97%
B	Open access	485	4.61%
	Other	10030	95.39%

**Table 2 materials-11-02530-t002:** Subject areas found in query A.

Subject Area	Number of Publications (NP)
Materials Science	195
Engineering	182
Biochemistry, Genetics, and Molecular Biology	104
Chemical Engineering	77
Physics and Astronomy	35
Medicine	34
Chemistry	22
Dentistry	7
Pharmacology, Toxicology, and Pharmaceutics	6
Agricultural and Biological Sciences	4
Immunology and Microbiology	4
Business, Management, and Accounting	2
Computer Science	2
Energy	1
Health Professions	1
Neuroscience	1

**Table 3 materials-11-02530-t003:** Subject areas found in query B.

Subject Areas	Number of Publications (NP)
Engineering	5531
Materials Science	5245
Biochemistry, Genetics, and Molecular Biology	4190
Medicine	2983
Chemical Engineering	2957
Chemistry	925
Physics and Astronomy	801
Pharmacology, Toxicology, and Pharmaceutics	719
Dentistry	334
Immunology and Microbiology	305
Neuroscience	165
Computer Science	154
Agricultural and Biological Sciences	150
Multidisciplinary	96
Health Professions	73
Environmental Science	61
Mathematics	48
Social Sciences	26
Energy	21
Veterinary	17
Arts and Humanities	16
Undefined	14
Nursing	12
Business, Management, and Accounting	10
Earth and Planetary Sciences	7
Decision Sciences	1

**Table 4 materials-11-02530-t004:** Types of documents.

Document Type	Number of Documents		Percentage of Documents	
	A Query	B Query	A Query	B Query
Articles	207	7749	78.41%	73.69%
Conference Papers	29	601	10.98%	5.72%
Reviews	19	1677	7.20%	15.95%
Other	9	488	3.41%	4.64%

**Table 5 materials-11-02530-t005:** Source titles with bucketing regarding the number of publications for A query.

Source Title (ST)	NP Per ST (NP_ST_)	TNP ∑(NP_ST_ × ST)	Overall Percentage
Journal of Biomedical Materials Research Part A	25	25	9.47%
Journal of Materials Science: Materials in Medicine	24	24	9.09%
Acta Biomaterialia	21	21	7.95%
Biomaterials	17	17	6.44%
Materials Science and Engineering C	13	13	4.92%
Biomedical Materials Bristol; Ceramics International; Journal of Biomedical Materials Research Part B Applied Biomaterials; Journal of Materials Chemistry B; Key Engineering Materials	5	25	9.47%
Journal of Clinical Rehabilitative Tissue Eng. Research	4	4	1.52%
Advanced Engineering Materials; Advanced Materials Research; Biomacromolecules; Ceramic Transactions; Journal of Biomaterials Applications; Journal of the Mechanical Behavior of Biomedical Materials; Kuei Suan Jen Hsueh Pao Journal of the Chinese Ceramic Society; PloS One; Tissue Engineering Part A	3	27	10.23%
ACS Applied Materials and Interfaces; Advances in Applied Ceramics; Biofabrication; Biomed Research International; Colloids and Surfaces B Biointerfaces; Dental Materials; Expert Review of Medical Devices; Gongneng Cailiao Journal of Functional Materials; IET Nanobiotechnology; International Journal of Artificial Organs; International Journal of Biological Macromolecules; Journal of Biomaterials and Tissue Engineering; Journal of Biomedical Materials Research; Materials Research Society Symposium Proceedings; Materials Science Forum; Materialwissenschaft und Werkstofftechnik; RSC Advances; Scientia Iranica	2	36	13.64%
2014 21st Iranian Conference on Biomedical Engineering (ICBME) 2014; American Association of Pharmaceutical Scientists (AAPS) PharmSciTech; Advanced Drug Delivery Reviews; Advanced Functional Materials; Advanced Structured Materials; Advances in Experimental Medicine and Biology; Annals of Biomedical Engineering; Antimicrobial Agents and Chemotherapy; Artificial Cells Nanomedicine and Biotechnology; Bio Medical Materials and Engineering; Biochemical and Biophysical Research Communications; Biointerphases; Biological and Pharmaceutical Bulletin; Biomedical Materials; Bone Regeneration Growth Factors Augmentation Procedures and Tissue Engineering Applications; Cell Proliferation; Ceramic Engineering and Science Proceedings; Chinese Journal of Tissue Engineering Research; Chinese Medical Journal; Composites Science and Technology; Connective Tissue Research; Dental Clinics of North America; Dental Research Journal; Energy Procedia; European Journal of Plastic Surgery; Faraday Discussions; Frontiers in Bioengineering and Biotechnology; Frontiers in Bioscience Elite; High Value Manufacturing Advanced Research in Virtual and Rapid Prototyping Proceedings of the 6th International Conference on Advanced Research and Rapid Prototyping Vr@p 2013; Injury; Interface Focus; International Journal of Applied Ceramic Technology; International Journal of Materials and Product Technology; International Journal of Molecular Sciences; International Journal of Nanomedicine; International Journal of Polymeric Materials and Polymeric Biomaterials; Journal of Biomechanics; Journal of Ceramic Science and Technology; Journal of Clinical Periodontology; Journal of Contemporary Dental Practice; Journal of Controlled Release; Journal of Hand Surgery; Journal of Materials Science; Journal of Non Crystalline Solids; Journal of Porous Materials; Journal of Prosthodontics; Journal of Sol Gel Science and Technology; Journal of the European Ceramic Society; Journal of Tissue Engineering and Regenerative Medicine; Korean Journal of Materials Research; Langmuir; Macromolecular Bioscience; Materials and Design; Materials Chemistry and Physics; Materials Letters; Materials Research; Materials Science and Engineering A; Materials Science and Technology Conference and Exhibition 2015 MS and T 2015; Microporous and Mesoporous Materials; Molecular Medicine Reports; New Developments in Cell Research; Orthopade; Pan American Health Care Exchanges Pahce; Procedia Engineering; Proceedings of The Institution of Mechanical Engineers Part H Journal of Engineering in Medicine; Progress in Polymer Science; Recent Patents on Regenerative Medicine; Sheng Wu Yi Xue Gong Cheng Xue Za Zhi Journal of Biomedical Engineering Shengwu Yixue Gongchengxue Zazhi; Society of Plastics Engineers Eurotech 2013; Tissue Engineering; Virtual Prototyping and Bio Manufacturing in Medical Applications; Wuji Cailiao Xuebao Journal of Inorganic Materials	1	72	27.27%

**Table 6 materials-11-02530-t006:** Affiliations organised after bucketing by number of publications.

Position	Affiliation	NP Per Affiliation Group
1	Imperial College of London	67
2	Friedrich-Alexander-Universität Erlangen-Nürnberg	29
3	Universite de Liege	16
4	Ministry of Education China	15
5	South China University of Technology	13
6	University College London (UCL) Eastman Dental Institute	12
7	King’s College London	11
8	Chinese Academy of Sciences; Shanghai Institute of Ceramics Chinese Academy of Sciences	10
9	Isfahan University of Medical Sciences; Shanghai Ninth People’s Hospital	8
10	McGill University	7
11	Queensland University of Technology (QUT); UCL; Tehran University of Medical Sciences; Shanghai Jiao Tong University School of Medicine; Consorcio Centro de Investigación Biomédica en Red de Bioingeniería, Biomateriales y Nanomedicina (CIBER-BBN)	6
12	University of Westminster; University of Manchester; University of Warwick; Shanghai Jiao Tong University; Tongji University	5
13	Universita degli Studi di Modena e Reggio Emilia; Politecnico di Torino; Tianjin University; Universidade de Aveiro	4
14	Pasteur Institute of Iran; Florida Institute of Technology; Soochow University; East China University of Science and Technology; Changzheng Hospital; Universidade Federal de Sao Carlos; Centro de Investigacao em Materiais Ceramicos e Compositos; National Research Centre; Xi’an Jiaotong University; Sun Yat-Sen University; Queen Mary, University of London; Missouri University of Science and Technology; Universidad Nacional de Salta; Islamic Azad University, Najafabad Branch; Isfahan University of Technology; University of Cambridge; Universidad Michoacana de San Nicolas de Hidalgo	3
15	Mo-Sci Corporation; Nanoforce Technology Limited; Universite Blaise Pascal; Universidad de Buenos Aires; University of Sheffield; National Institute for Materials Science Tsukuba; Chelsea and Westminster Hospital; Southern Medical University; Instituto de Investigaciones en Ciencia y Tecnologia de Materiales; Politechnika Warszawska; Zhejiang University; Consejo Nacional de Investigaciones Cientificas y Tecnicas; University of Pennsylvania; The University of Hong Kong; Lawrence Berkeley National Laboratory; Amirkabir University of Technology; Amrita Institute of Medical Sciences India; Universidade de Sao Paulo—USP; Harbin Normal University; University of Madras; Tampere University of Technology; University of Newcastle, Australia; Universitat Politècnica de València; Pukyong National University; University of Leeds; Iran University of Science and Technology; Polish Academy of Sciences; Universidade Federal de Sao Paulo; Shahrekord University; Aristotle University of Thessaloniki; Institut National des Sciences Appliquees de Lyon; Rheinisch-Westfalische Technische Hochschule Aachen; AGH University of Science and Technology; Monash University; Universidade do Minho; Comision Nacional de Energia Atomica Argentina; Uniwersytet Jagiellonski w Krakowie; Istanbul Teknik Universitesi; Shanghai Normal University; St. Mark’s Hospital and Academic Institute; University of Tehran; Semmelweis Egyetem; Materials and Energy Research Centre Iran; Second Military Medical University; The University of Sydney; Universidad Complutense de Madrid; Polytechnic Institute of Leiria; Universita di Pisa; University of Malaya; Wuhan University; Northwick Park Hospital; Tarbiat Modares University; University of Belgrade; Xijing Hospital; Changhai Hospital; State Key Laboratory of Bioreactor Engineering Shanghai; The Fourth Military Medical University; Zhejiang California International NanoSystems Institute; Stem Cell Technology Research Center; Vysoke Uceni Technicke v Brne, Stredoevropsku Technologicku Institut; Amrita Vishwa Vidyapeetham University, Kochi	2
16	First People’s Hospital of Changzhou; OsteoBiologics, Inc.; China National Academy of Nanotechnology and Engineering; School of Mechanical and Automotive Engineering; CJD Pharmacology Consultants; NanotecMARIN GmbH; Laboratory of Chemistry and Biomaterials; Research Institute of Polymeric Materials; GE Sensing and Inspection Technologies GmbH; Lima-Lto S.p.A.; Federal University Technological Paraná; National Science Research Council; Center for Tissue Engineering C.I.T.; Institute for Biotechnology and Bioengineering (IBB); Advanced Materials Research Centre AMREC; University of Natural Resources and Life Science; ICVS/3B’s-PT Government Associate Laboratory; People’s Liberation Army 81 Hospital; Imam Hossein Comprehensive University; Instituto de Tecnologías y Ciencias de la Ingeniería “Hilario Fernández Long” (INTECIN) of the University of Buenos Aires (INTECIN UBA-CONICET); Midland Regional Hospital at Tullamore (MRHT); Rui’An People’s Hospital and the Third Affiliated Hospital to Wenzhou Medical University; Donostia University Hospital Osakidetza-Basque Health Service and BIODONOSTIA; FEI Visualization Sciences Group; Munzur University; Third Affiliated Hospital to Wenzhou Medical University; Universidad de Murcia; University of Connecticut Health Center; Nagoya University; Urmia University; National Institute of Technology Rourkela; Medizinische Fakultat und Universitats Klinikum Aachen; Thapar University; Universidade Federal do ABC, Santo André, Sao Paulo, Brazil; Gap; Universitat de Barcelona; Lackland Air Force Base; Radboud University Nijmegen Medical Centre; Future University in Egypt; Dankook University; Research Institute of Petroleum Industry, Tehran; Ohio State University; University of Mississippi Medical Center; Panepistimion Ioanninon; Gottfried Wilhelm Leibniz Universitat; Universita Campus Bio-Medico di Roma; Nanyang Technological University; University of Iowa College of Dentistry; UNESP–Universidade Estadual Paulista; University of Missouri-Columbia; Newcastle University, United Kingdom; University of Missouri-Kansas City; Universitat Politecnica de Catalunya; Prairie View A and M University; Universitat Leipzig; Wenzhou Medical University	1

**Table 7 materials-11-02530-t007:** Publications per country, population in 2017, gross domestic product (GDP) (reference to 2016, in 10^12^ USD), number of publications (NP) per million of habitants, and the ratio of number of publications per respective GDP.

Position	Country	NP	Population (2017) (10^6^)	GDP (2016) (10^12^ USD)	NP/10^6^ Habitants	NP/GDP (10^12^ USD)
1	United Kingdom	84	66.1816	2.6479	1.2692	31.7233
2	China	58	1410.0000	11.2183	0.0411	5.1701
3	Germany	37	82.1142	3.4778	0.4506	10.6389
4	United States	31	66.1816	18.6245	0.4684	1.6645
5	Belgium	17	11.4293	0.4680	1.4874	36.3283
6	Iran	17	81.1628	0.9323	0.2095	18.2353
7	Spain	13	46.3543	1.2373	0.2804	10.5071
8	Australia	12	24.4506	1.3045	0.4908	9.1992
9	Canada	11	36.6242	1.5298	0.3003	7.1907
10	Italy	11	59.3599	1.8589	0.1853	5.9174
11	India	9	1339.0000	2.2596	0.0067	3.9829
12	France	7	64.9795	2.4655	0.1077	2.8392
13	Brazil	6	209.2883	1.7959	0.0287	3.3409
14	Poland	6	38.1707	0.4714	0.1572	12.7280
15	Portugal	6	10.3295	0.2048	0.5809	29.2916
16	Argentina	4	44.2710	0.5459	0.0904	7.3278
17	Egypt	4	97.5532	0.2701	0.0410	14.8069
18	Japan	4	127.4845	4.9362	0.0314	0.8103
19	Malaysia	4	31.6243	0.2965	0.1265	13.4893
20	South Korea	4	50.9822	3.5730	0.0785	1.1195
21	Finland	3	5.5232	0.2385	0.5432	12.5785
22	Mexico	3	129.1633	1.0769	0.0232	2.7857
23	Turkey	3	80.7450	0.8637	0.0372	3.4734
24	Austria	2	8.7355	0.3908	0.2290	5.1177
25	Czech Republic	2	10.6183	0.1953	0.1884	10.2404
26	Greece	2	11.1598	0.1927	0.1792	10.3793
27	Hong Kong	2	7.3649	0.0215	0.2716	92.9501
28	Hungary	2	9.7216	0.1258	0.2057	15.8961
29	Ireland	2	4.7617	0.3048	0.4200	6.5613
30	Serbia	2	8.7906	0.0383	0.2275	52.2195
31	Singapore	2	5.7088	0.2969	0.3503	6.7352
32	Switzerland	2	8.4760	0.6689	0.2360	2.9902
33	Colombia	1	49.0656	0.2825	0.0204	3.5403
34	Morocco	1	35.7396	0.1036	0.0280	9.6519
35	Netherlands	1	17.0359	0.7772	0.0587	1.2866
36	Pakistan	1	197.0160	0.2825	0.0051	3.5397
37	Russian Federation	1	143.9898	1.2460	0.0069	0.8026
38	Taiwan	1	23.6265	0.0222	0.0423	45.1201
39	Undefined	6	-	-	-	-

**Table 8 materials-11-02530-t008:** Country positions after sorting out the data according to the ratios of ‘Number of publications/Million habitants (left); and the ‘Number of publications/GDP (GDP in 10^12^ USD) (right).

Position	Country	NP/10^6^ Habitants		Country	NP/GDP (10^12^ USD)
1	Belgium	1.4874	1	Hong Kong	92.9501
2	United Kingdom	1.2692	2	Serbia	52.2195
3	Portugal	0.5809	3	Taiwan	45.1201
4	Finland	0.5432	4	Belgium	36.3283
5	Australia	0.4908	5	United Kingdom	31.7233
6	United States	0.4684	6	Portugal	29.2916
7	Germany	0.4506	7	Iran	18.2353
8	Ireland	0.4200	8	Hungary	15.8961
9	Singapore	0.3503	9	Egypt	14.8069
10	Canada	0.3003	10	Malaysia	13.4893
11	Spain	0.2804	11	Poland	12.7280
12	Hong Kong	0.2716	12	Finland	12.5785
13	Switzerland	0.2360	13	Germany	10.6389
14	Austria	0.2290	14	Spain	10.5071
15	Serbia	0.2275	15	Greece	10.3793
16	Iran	0.2095	16	Czech Republic	10.2404
17	Hungary	0.2057	17	Morocco	9.6519
18	Czech Republic	0.1884	18	Australia	9.1992
19	Italy	0.1853	19	Argentina	7.3278
20	Greece	0.1792	20	Canada	7.1907
21	Poland	0.1572	21	Singapore	6.7352
22	Malaysia	0.1265	22	Ireland	6.5613
23	France	0.1077	23	Italy	5.9174
24	Argentina	0.0904	24	China	5.1701
25	South Korea	0.0785	25	Austria	5.1177
26	Netherlands	0.0587	26	India	3.9829
27	Taiwan	0.0423	27	Colombia	3.5403
28	China	0.0411	28	Pakistan	3.5397
29	Egypt	0.0410	29	Turkey	3.4734
30	Turkey	0.0372	30	Brazil	3.3409
31	Japan	0.0314	31	Switzerland	2.9902
32	Brazil	0.0287	32	France	2.8392
33	Morocco	0.0280	33	Mexico	2.7857
34	Mexico	0.0232	34	United States	1.6645
35	Colombia	0.0204	35	Netherlands	1.2866
36	Russian Federation	0.0069	36	South Korea	1.1195
37	India	0.0067	37	Japan	0.8103
38	Pakistan	0.0051	38	Russian Federation	0.8026

**Table 9 materials-11-02530-t009:** Batch compositions of the glasses within the fluorapatite (FA)–diopside (Di) joint (mol %).

Glass	CaO	MgO	SiO_2_	P_2_O_5_	CaF_2_
FA-100	69.23	0.00	0.00	23.08	7.69
FA-90	63.15	3.43	6.87	19.91	6.64
FA-80	57.57	6.59	13.19	16.99	5.66
FA-70	52.40	9.51	19.03	14.30	4.77
FA-60	47.62	12.21	24.44	11.80	3.93
FA-50	43.18	14.73	29.45	9.48	3.16
FA-40	39.04	17.07	34.12	7.33	2.45
FA-35	37.07	18.18	36.34	6.30	2.10
FA-30	35.18	19.25	38.49	5.31	1.77
FA-25	33.34	20.28	40.57	4.36	1.45
FA-20	31.57	21.29	42.57	3.43	1.14
FA-10	28.18	23.20	46.41	1.66	0.55
FA-0	25.00	25.00	50.00	0.00	0.00

**Table 10 materials-11-02530-t010:** Batch compositions of the glasses within Di–FA–wollastonite (W) ternary system (mol %).

Glass	CaO	MgO	SiO_2_	P_2_O_5_	CaF_2_
W-10	34.11	18.76	42.52	3.45	1.15
W-20	36.70	16.20	42.47	3.48	1.16
W-30	39.32	13.60	42.41	3.50	1.17
W-40	41.98	10.96	42.35	3.53	1.18
W-50	44.68	8.28	42.30	3.56	1.19
W-60	47.42	5.56	42.24	3.58	1.19
W-70	50.20	2.80	42.18	3.61	1.20
W-80	53.03	0.00	42.12	3.64	1.21

**Table 11 materials-11-02530-t011:** Batch compositions of the glasses within Di–FA– tricalcium phosphate (TCP) ternary system (mol %).

Glass	Formula	CaO	MgO	SiO_2_	P_2_O_5_	CaF_2_
TCP-10	80Di-10FA-10TCP	32.00	21.29	42.57	3.57	0.57
TCP-20	70Di-10FA-20TCP	36.07	19.24	38.49	5.61	0.59
TCP-30	60Di-10FA-30TCP	40.42	17.06	34.12	7.79	0.61
TCP-40	50Di-10FA-40TCP	45.08	14.72	29.44	10.12	0.63

**Table 12 materials-11-02530-t012:** Batch compositions of the glasses (mol %).

Glass	CaO	MgO	SiO_2_	P_2_O_5_	ZnO	SrO	CaF_2_
ZS-0	36.07	19.24	38.49	5.61	0.00	0.00	0.59
ZS-2	34.07	17.24	38.49	5.61	2.00	2.00	0.59
ZS-4	32.07	15.24	38.49	5.61	4.00	4.00	0.59
ZS-6	30.07	13.24	38.49	5.61	6.00	6.00	0.59
ZS-8	28.07	11.24	38.49	5.61	8.00	8.00	0.59
ZS-10	26.07	9.24	38.49	5.61	10.00	10.00	0.59

**Table 13 materials-11-02530-t013:** Batch compositions of the glasses (mol %).

Glass	CaO	MgO	SiO_2_	P_2_O_5_
Di-0	75.00	0.00	0.00	25.00
Di-10	68.13	3.43	6.87	21.57
Di-20	61.82	6.59	13.18	18.41
Di-30	55.99	9.50	19.02	15.49
Di-40	50.57	12.22	24.42	12.79
Di-50	45.55	14.73	29.44	10.28
Di-60	40.88	17.06	34.12	7.94
Di-70	36.52	19.24	38.48	5.76
Di-80	32.44	21.28	42.57	3.71
Di-90	28.60	23.20	46.40	1.08
Di-100	25.00	25.00	50.00	0.00

**Table 14 materials-11-02530-t014:** Bioactive glasses compositions (mol %).

Bioactive	SiO_2_	P_2_O_5_	CaO	Na_2_O	MgO	ZnO	SrO	CaF_2_
FastOs^®^BG	38.49	5.61	36.07	0.00	19.24	0.00	0.00	0.59
ZS-2	38.49	5.61	34.07	0.00	17.24	2.00	2.00	0.59
ZS-4	38.49	5.61	32.07	0.00	15.24	4.00	4.00	0.59
Di-70	38.48	5.76	36.52	0.00	19.24	0.00	0.00	0.00
Di-80	42.57	3.71	32.44	0.00	21.28	0.00	0.00	0.00
45S5 Bioglass^®^	46.10	2.60	26.9	24.4	0.00	0.00	0.00	0.00

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
