# Peer review of "Bioactive Glasses and Glass-Ceramics for Healthcare Applications in Bone Regeneration and Tissue Engineering"

_materials, 2018, doi:10.3390/ma11122530_

Round 1

Reviewer 1 Report

This manuscript focuses on bioactive glasses and glass-ceramics and presents a detailed summary of events from their development in 1960s until today. The authors did a great job writing this excellent manuscript and collecting all the information however they did not a great job citing their work. There is a large number of sentences/paragraphs that are not cited and they are not the work of the authors. Some examples are listed below in the comments. The authors are recommended to go through the whole article and make sure that everything (except for their own ideas) is cited properly. 

Additionally the authors do not recognize the work done by some well known groups in the field of bioglass and glass-ceramics. For examples, Dr. Mark Towler (Ryerson University, Canada) is a well known professor in the field of bioglass and has over 150 publications on bioglass. Dr. Towler has patents for bioglass systems. He was the first to replace Aluminum with Zinc which was a significant contribution in the field. He was also the first to add tantalum to bioglasses which made them potential materials for orthopedic applications. This is a review paper and therefore the authors are expected to cover all research that is happening in the field and recognize the work done by all researchers in the field. The authors are recommended to look more into research happening into the field and present up-to-date examples of bioactive glass and glass-ceramic advances and how they contributed to the development of medical innovations. 

Following are some specific comments however as mentioned earlier the authors are recommended to go through the whole review and fix similar issues. 

Line 61: Change 'structured' to structure .

Figure 1: Authors did not identify what does letter 'R' refer to ! 

Line 130: "(often not welcome in bioactive glasses)" the authors did not explain why aluminum was not welcome in bioglasses and how it was removed and what did researchers replace it with. 

Lines 131-141: No citations. Please cite your work. 

Lines 143-151: there is only 2 citations for the whole paragraph. Authors should cite their work. 

Lines 157-162: add references. 

Line 167: Mention some examples of the applications 

Lines 168-176: Add references

Line 178: change 'based in' to based on. 

Line 177: what are clinical/industrial examples of the mixed glasses. The authors provide very short paragraph on such a big topic. The authors need to provide more information to clear the picture. 

Line 185: 'specific applications such as biomedical uses' such as what? 

Line 194: what is 'viz.' ?? 

Line 239: 'because they derived from a parent glass' add are between they and derived 

Lines 250-267: no citations!! 

Lines 269-272: long sentence and is not clear. Please re-phrase. 

Line 315: mention some of the attractive properties. 

I would like to congratulate the authors on their hard work however I believe the paper need some work before it can be accepted for publication. 

Author Response

Letter of response to Reviewer#1

01. This manuscript focuses on bioactive glasses and glass-ceramics and presents a detailed summary of events from their development in 1960s until today. The authors did a great job writing this excellent manuscript and collecting all the information however they did not a great job citing their work. There is a large number of sentences/paragraphs that are not cited and they are not the work of the authors. Some examples are listed below in the comments. The authors are recommended to go through the whole article and make sure that everything (except for their own ideas) is cited properly. 

Answer: Thanks for the overall positive appraisal on our review article, and also for the remarks relative to aspects that need further attention. The manuscript was carefully revised to cope with all the pertinent remarks in order to improve the clarity and the quality of the manuscript.

02. Additionally the authors do not recognize the work done by some well known groups in the field of bioglass and glass-ceramics. For examples, Dr. Mark Towler (Ryerson University, Canada) is a well known professor in the field of bioglass and has over 150 publications on bioglass. Dr. Towler has patents for bioglass systems. He was the first to replace Aluminum with Zinc which was a significant contribution in the field. He was also the first to add tantalum to bioglasses which made them potential materials for orthopedic applications. This is a review paper and therefore the authors are expected to cover all research that is happening in the field and recognize the work done by all researchers in the field. The authors are recommended to look more into research happening into the field and present up-to-date examples of bioactive glass and glass-ceramic advances and how they contributed to the development of medical innovations. 

Answer: Thanks for calling our attention to the contributions to the field by Dr. Mark Towler and co-workers. The revised manuscript now includes references to the most relevant related findings: (i) penultimate paragraph of section “3. Bioactive glasses and glass-ceramics”; (ii) Section “3.4.1. Ion-doped bioactive glasses”, points 4, 10 and 11.

03. Following are some specific comments however as mentioned earlier the authors are recommended to go through the whole review and fix similar issues. 

Line 61: Change 'structured' to structure .

Figure 1: Authors did not identify what does letter 'R' refer to ! 

Line 130: "(often not welcome in bioactive glasses)" the authors did not explain why aluminum was not welcome in bioglasses and how it was removed and what did researchers replace it with. 

Lines 131-141: No citations. Please cite your work. 

Lines 143-151: there is only 2 citations for the whole paragraph. Authors should cite their work. 

Lines 157-162: add references. 

Line 167: Mention some examples of the applications 

Lines 168-176: Add references

Line 178: change 'based in' to based on. 

Line 177: what are clinical/industrial examples of the mixed glasses. The authors provide very short paragraph on such a big topic. The authors need to provide more information to clear the picture. 

Line 185: 'specific applications such as biomedical uses' such as what? 

Line 194: what is 'viz.' ?? 

Line 239: 'because they derived from a parent glass' add are between they and derived 

Lines 250-267: no citations!! 

Lines 269-272: long sentence and is not clear. Please re-phrase. 

Line 315: mention some of the attractive properties. 

Answer: Thank you so much for your detailed reading and for pointing out the passages that need specific improvements. All of them have been addressed in the revised manuscript.

04. I would like to congratulate the authors on their hard work however I believe the paper need some work before it can be accepted for publication. 

Answer: Thanks again for the positive appraisal. The manuscript was carefully revised to cope with all the pertinent remarks in order to improve the clarity and the quality of the manuscript.

Reviewer 2 Report

In this review the authors presented the knowledge-based tools towards guiding young researchers in the design of new bioactive glass compositions taking into account the desired functional properties. The presented materials are very useful and recommend it for publication after the following minor points are addressed:

- The schematic representation of a single SiO4 tetrahedron and structural units that build up a phosphate and borate based glasses would improve the review.

- I recommend the schematic illustration of apatite layers function on bioactive glasses in body fluid

- Please indicate in the table 1 the A and B meaning

Author Response

Letter of response to Reviewer #2

01. In this review the authors presented the knowledge-based tools towards guiding young researchers in the design of new bioactive glass compositions taking into account the desired functional properties. The presented materials are very useful and recommend it for publication after the following minor points are addressed:

- The schematic representation of a single SiO4 tetrahedron and structural units that build up a phosphate and borate based glasses would improve the review.

- I recommend the schematic illustration of apatite layers function on bioactive glasses in body fluid

- Please indicate in the table 1 the A and B meaning. 

Answer: The authors are thankful for the comments and suggestions received from the Reviewer, which were fully addressed and helped to improve the quality and completeness of the review.

Reviewer 3 Report

the review deals on the bioglass and bioglass-ceramics used in bone tissue engineering. the review is very well written and I appreciate all the statistics and tables that summarise or highlight the main aspects of the field. 

Just a criticism: i found very long the initial section on the general overview of bioglasses. my suggestion is too shorten it drastically due to the fact the review is very long.

This one can give more space for describing other important aspects of the bioglass applied in bone tissue engineering, such as polymeric coating on bioglass ceramics in order to mimick the bone composition:

Biomimetic coating on bioactive glass-derived scaffolds mimicking bone tissue Journal of Biomedical Materials Research - Part A 100 A(12), pp. 3259-3266 2012

Author Response

Letter of response to Reviewer #3

01. The review deals on the bioglass and bioglass-ceramics used in bone tissue engineering. the review is very well written and I appreciate all the statistics and tables that summarise or highlight the main aspects of the field.

Answer: Thanks for your overall positive appraisal.

02. Just a criticism: i found very long the initial section on the general overview of bioglasses. my suggestion is too shorten it drastically due to the fact the review is very long.

Answer: The authors respectful disagree with the Reviewer in this particular comment. As well pointed out by Reviewer 2, the authors aimed at providing the knowledge-based tools towards guiding young researchers in the design of new bioactive glass compositions taking into account the desired functional properties. In our opinion, the size of the initial section is appropriate for conveying a comprehensive account on the relevant related concepts.

03. This one can give more space for describing other important aspects of the bioglass applied in bone tissue engineering, such as polymeric coating on bioglass ceramics in order to mimick the bone composition:

Biomimetic coating on bioactive glass-derived scaffolds mimicking bone tissue Journal of Biomedical Materials Research - Part A 100 A(12), pp. 3259-3266 2012

Answer: Thanks for the suggestion. The progresses on polymer coating on bioglass scaffolds has been now included in the revised version of our manuscript, redirecting the readers’ attention towards an excellent, recent and comprehensive topical review (DOI: 10.1586/17434440.2015.958075), the article indicated by the reviewer, as well as other works which we found of interest. Please see the new paragraph at page 11.

Round 2

Reviewer 1 Report

The authors have addressed all my concerns. I recommend accepting it for publication.